# Validating Mechanistic Interpretations: An Axiomatic Approach

**Nils Palumbo** [1]  **Ravi Mangal** [2]  **Zifan Wang** [3]  **Saranya Vijayakumar** [4]  **Corina Păsăreanu** [4]  **Somesh Jha** [1]

## Abstract

Mechanistic interpretability aims to reverse engineer the computation performed by a neural network in terms of its internal components. Although there is a growing body of research on mechanistic interpretation of neural networks, the notion of a *mechanistic interpretation* itself is often ad-hoc. Inspired by the notion of abstract interpretation from the program analysis literature that aims to develop approximate semantics for programs, we give a set of axioms that formally characterize a mechanistic interpretation as a description that approximately captures the semantics of the neural network under analysis in a compositional manner. We demonstrate the applicability of these axioms for validating mechanistic interpretations on an existing, well-known interpretability study as well as on a new case study involving a Transformer-based model trained to solve the well-known 2-SAT problem.

## 1. Introduction

Neural networks are notoriously opaque. Mechanistic interpretability seeks to reverse-engineer the computation described by a neural network in a human-interpretable form. Typically, the resulting *mechanistic interpretation* takes the form of a circuit (i.e., program) operating over features (i.e., symbols). The features refer to human-interpretable properties of the input (such as edges, textures, or shapes for vision models and part-of-speech tags, named entities, or semantic relationships for language models) that the model represents internally in its representations spaces whereas the circuit refers to a chain of human-interpretable operations, encoded by the model in its layers, such that each operation processes and transforms the features (Millière & Buckner, 2024). While mechanistic interpretability has been used to analyze

aspects of vision as well as language models, it has found most success in analyzing small models trained to solve specific algorithmic tasks such as modular addition (Nanda et al., 2023a). Although such small models are not necessarily of practical use, analyzing them has been fruitful for developing the techniques needed to mechanistically interpret.

Despite the broad interest in mechanistic interpretability, we observe that the notion of a *mechanistic interpretation* has so far been primarily defined in an ad-hoc manner. In this work, we propose a collection of axioms that characterize a mechanistic interpretation.[1] Our view on mechanistic interpretability is inspired by the notion of *abstract interpretation* from the program analysis literature (Cousot & Cousot, 1977; 1992) that seeks to approximate the semantics of programs in order to make their analysis computationally feasible. Similarly, our axioms are meant to capture the property that the mechanistic interpretation should capture the semantics of the model. Further, our axioms also capture the intuition that such an approximation should respect the compositional structure of the neural network—not only should the input-output behavior of the mechanistic interpretation be as close as possible to the original model, but every step in the reverse-engineered algorithm should be as close as possible to the computation performed by the internal components of the model (i.e., neurons and layers). Our axioms clarify the responsibilities of an analyst; in addition to providing a mechanistic interpretation, an analyst also needs to present evidence to support the fact that the provided interpretation is valid, i.e., satisfies these axioms. Note that our axioms present a general framework for validating mechanistic interpretations. Closest to our work are recent papers by Geiger et al. (2021) and Chan et al. (2022) that seek to formalize the notion of mechanistic interpretability from a causal perspective. We believe that these formulations are complementary to our axioms.

We demonstrate the applicability of our axioms for validating mechanistic interpretations via two case studies[2]—the existing, well-known analysis by Nanda et al. (2023a) of a model with a Transformer-based architecture (Vaswani et al., 2017) trained to solve modular addition, and a new analysis

[1]University of Wisconsin-Madison [2]Colorado State University [3]Scale AI [4]Carnegie Mellon University. Correspondence to: Nils Palumbo <npalumbo@wisc.edu>, Ravi Mangal <ravi.mangal@colostate.edu>.

*Proceedings of the 42nd International Conference on Machine Learning*, Vancouver, Canada. PMLR 267, 2025. Copyright 2025 by the author(s).

---

[1]In the same sense that group axioms characterize the notion of an abstract group in abstract algebra.

[2]Our implementation is available at https://github.com/nilspalumbo/axiomatic-validation.

of a small Transformer-based model trained by us to solve a Boolean satisfiability problem. For the new case study, we choose Boolean satisfiability as a challenging yet well-understood problem that serves to highlight new techniques and our axiomatic evaluation criteria for mechanistic interpretability. Further, there are many other problems in computer science that reduce to it, for example, graph reachability is reducible to 2-SAT. To keep the analysis tractable, we focus on the 2-SAT problem with a fixed number of clauses and variables; the problem can be solved in polynomial time (while the more general 3-SAT problem is NP-complete).

To apply our axioms, we first need to extract a mechanistic interpretation of the model under analysis. Although such an interpretation is already known for the modular arithmetic model, we present a new analysis for the 2-SAT model that is guided by our axioms. Through our analysis, we are able to reverse engineer the algorithm learned by the 2-SAT model—the model parses the formula into a clause-level representation in its initial layers and then uses the later layers to evaluate the formula satisfiability via enumeration of different possible valuations of the Boolean variables. Since we only consider 2-SAT formulas with ten clauses and five variables, such an enumerative approach is computationally feasible and our model has the capacity to encode it. We use novel variants of attention pattern analysis to interpret the first Transformer block of the model. For the second (i.e., final) Transformer block, attention pattern analysis does not suffice and we use automated learning of functions (decision trees) to derive an interpretation. Finally, for both the case studies, we present evidence that the mechanistic interpretations extracted for these models indeed satisfy our axioms.

## 2. Related Work

Work on interpreting neural networks can be divided between techniques that find input features with the largest effect on model behavior (*Input Interpretability*) such as gradient-based (Simonyan et al., 2013; Sundararajan et al., 2017; Leino et al., 2018; Smilkov et al., 2017) and activation-based (Olah et al., 2017; Petsiuk et al., 2018; Fong & Vedaldi, 2017; Selvaraju et al., 2019; Wang et al., 2020) attributions, and those that interpret the internal reasoning performed by a model (*Internal Interpretability*). We focus on the latter.

Mechanistic interpretability typically refers to analyses that seek to understand the model behavior *completely*, i.e., they seek to reverse engineer, in a human-understandable, and hence simplified form, the full algorithm that the neural network learns. Such analyses have tended to focus on analyzing toy models trained for algorithmic tasks such as modular addition (Nanda et al., 2023a; Zhong et al., 2023), finding greatest common divisors (Charton, 2023), $n$-digit integer addition (Quirke & Barez, 2024), and finite group op-

erations (Chughtai et al., 2023). Mechanistic interpretability has also been used to refer to analyses that only seek to understand specific aspects of the model behavior. Such analyses, often referred to as *circuit analyses*, isolate circuits, i.e., subgraphs of the neural network computational graph, that are responsible for the behavior of interest (Cammarata et al., 2020; Olsson et al., 2022; Wang et al., 2023; Lepori et al., 2024; Wu et al., 2023b; Lieberum et al., 2023; Conmy et al., 2023). The circuits are validated by measuring the causal effect of ablating the circuit using techniques such as activation patching (Vig et al., 2020; Geiger et al., 2021; Heimersheim & Nanda, 2024) on a metric that measures the relevant model behavior. In either case, the standards for judging the validity of the mechanistic interpretations produced by such analyses are ad-hoc and do not take the compositional structure of the neural network into account. We axiomatize the standards for making such judgment in this paper.

Related to circuit analysis and the evaluation of mechanistic interpretations is emerging work on causal abstraction analysis (Geiger et al., 2021; 2024; Wu et al., 2023a;b; Chan et al., 2022) that extracts causal models to explain certain aspects of neural network behavior and uses techniques similar to activation patching for validating the causal abstractions.

The surveys by Millière & Buckner (2024), Mueller et al. (2024), and Räuker et al. (2023) are excellent resources for a broader and more detailed description of techniques for interpreting the internals of neural networks and the architectural components which form the interpreted units. Appendix A has a further discussion on internal interpretability techniques such as probing and attention pattern analysis.

## 3. Mechanistic Interpretation Axioms

Neural networks can be seen as programs in a purely functional language, which we denote $\lambda_T$, with basic operations corresponding to the commonly used neural network *layers* such as $Embed$, $Unembed$, $Lin$ (i.e., linear), $ReLU$, $Self\text{-}Attention$, $Convolution$, $Residual$, and so on. Since $\lambda_T$ is purely functional, all these operations are side-effect free—they simply transform inputs into outputs. The syntax of $\lambda_T$ is as follows ($x \in Var$):

$$t ::= Embed(x) \mid Unembed(x) \mid Lin(x) \mid ReLU(x) \mid$$
$$Self\text{-}Attention(x) \mid Convolution(x) \mid$$
$$Residual(x,t) \mid \ldots \mid t \circ t$$

The *decomposition* of a model $t \in \lambda_T$ is a list of programs in $\lambda_T$ such that the composition of these programs is syntactically equivalent to $t$. For example, consider the program $t := Lin \circ ReLU \circ Lin(x)$. One possible decomposition of $t$ is the list $[Lin, ReLU, Lin]$. Given a decomposition $d$, we use $d[i]$ to refer to the $i^{\text{th}}$ component of $d$, $d[:i]$ as a shorthand for $d[i-1] \circ d[i-2] \circ \ldots \circ d[1]$ and refer to it as a *prefix* of $t$ of length $i-1$ (with respect to $d$), and $d[i:]$

as the shorthand for $d[len(d)] \circ d[len(d) - 1] \circ \ldots \circ d[i]$ and referred to as a *suffix* of $t$.

**Mechanistic Interpretation.** The mechanistic interpretation of a neural network $t \in \lambda_T$ is a program in a different, purely functional language $\lambda_H$, that is human interpretable. The basic constructs of $\lambda_H$ will tend to be more abstract than the ones supported by $\lambda_T$; $\lambda_H$ might also be less expressive than $\lambda_T$ to aid human-interpretability. While the specific design of $\lambda_H$ and the question of how to judge whether a program is human-interpretable is domain-specific and subjective (and therefore difficult to formalize), we focus here on the question of how to judge whether a program $h \in \lambda_H$ is a valid mechanistic interpretation of the neural network $t \in \lambda_T$ under analysis.

At the very least, the input-output behavior of $h$ should be similar to $t$. Ideally, the input-output behaviors of the two programs should coincide on every input but this requirement is much too strong in practice. Assuming that the inputs are drawn from a distribution $\mathcal{D}$, we formalize the weaker requirement that the outputs of the two programs should be equal with a high probability as an axiom. For these equality conditions to be practically applicable, $h$ must operate over discrete values. Also, the output type of $t$ must be discrete, in particular, we assume that an argmax or top-k operator has been applied to the logits. We additionally require that the behaviors of $h$ and $t$ are equivalent at the level of individual components (obtained via suitably decomposing these programs) and that replacing a component of $t$ with the corresponding component of $h$ has a negligible effect on the neural network's output.

**Definition 3.1** (Mechanistic Interpretation). Given a model $t \in \lambda_T$ of type $X \to Y$ and a decomposition $d_t$ of $t$, an $\epsilon$-accurate mechanistic interpretation of $t$, with respect to decomposition $d_t$ and a distribution $\mathcal{D}$ over $X$, is a program $h \in \lambda_H$ with a decomposition $d_h$ such that $len(d_t) = len(d_h)$ and Axioms 1, 2, 3, and 4 hold.

Note that the definition is parametric in decomposition $d_t$ of $t$ as well as the input distribution $\mathcal{D}$. In practice, we will often consider layer or block-wise decompositions of $t$.

**Abstraction and Concretization.** The first four axioms use the notion of $\alpha$ and $\gamma$ functions which we describe here. The intuition is that a neural network $t$ and a corresponding mechanistic interpretation $h$ operate on different types of data representations. While $t$ operates over real-valued vectors, to aid interpretability, $h$ is intended to operate over human-interpretable features or symbols. Accordingly, $\alpha$ (or *abstraction*) functions map the real-valued activations (which we also refer to as *concrete representations*) computed by $t$ to the corresponding features or symbols (which we refer to as *abstract representations*) that $h$ operates over. Given a decomposition $d_t$ of $t$, we use $\alpha_i$ to refer to the $\alpha$ function mapping concrete representations computed by

$d_t[: i + 1]$, i.e., the prefix of length $i$. The $\alpha$ functions are similar to *probes* (Alain & Bengio, 2017) for extracting features from a model's internal activations. The $\gamma$ (or *concretization*) functions map in the opposite direction— they map abstract features or symbols operated on by $h$ to corresponding real-valued representations of those features in $t$'s representation space. The $\alpha$ and $\gamma$ functions in our axioms are directly inspired by the abstraction and concretization operations from the abstract interpretation literature used to map values between the original semantics and the abstract semantics of a program (Cousot & Cousot, 1977; 1992). We note that the $\alpha$ and $\gamma$ functions need to be individually instantiated every mechanistic interpretation.

**Axioms.** We define the following axioms for a particular choice of $\epsilon$; in our analysis we say that an interpretation satisfies an axiom with $\epsilon$ when the statement of that axiom holds.

Axiom 1 bounds the probability that the abstract and the concrete representations computed by the same-length prefix of $h$ and $t$, respectively, do not coincide (after applying $\alpha$ functions to map between the representations). Put differently, the mechanistic interpretation prefixes do not introduce too much error.

**Axiom 1** ($\epsilon$-Prefix Equivalence). $\forall i \in [len(d)]$.

$$\Pr_{x \sim \mathcal{D}}[\alpha_i \circ d_t[: i + 1](x) = d_h[: i + 1] \circ \alpha_0(x)] \geq 1 - \epsilon$$

In contrast, Axiom 2 requires that none of the individual components of the mechanistic interpretation $h$ introduce too much error. Axiom 2 does not imply Axiom 1 since the errors introduced by each component of the mechanistic interpretation can compound in the worst case (Dziri et al., 2023).

**Axiom 2** ($\epsilon$-Component Equivalence). $\forall i \in [len(d)]$.

$$\Pr_{x \sim \mathcal{D}}[\alpha_i \circ d_t[: i + 1](x) = d_h[i] \circ \alpha_{i-1} \circ d_t[: i](x)] \geq 1 - \epsilon$$

Axioms 3 and 4 are similar except that they consider equivalence of the output. Axiom 3 requires that replacing the prefixes of $t$ with the corresponding prefixes of $h$ (after applying appropriate $\alpha$ and $\gamma$ functions to map between the representations) has limited effect on $t$'s output. Axiom 4 requires the same when individual components of $t$ are replaced by corresponding components of $h$.

**Axiom 3** ($\epsilon$-Prefix Replaceability). $\forall i \in [len(d)]$.

$$\Pr_{x \sim \mathcal{D}}[t(x) = d_t[i + 1 :] \circ \gamma_i \circ d_h[: i + 1] \circ \alpha_0(x)] \geq 1 - \epsilon$$

**Axiom 4** ($\epsilon$-Component Replaceability). $\forall i \in [len(d)]$.

$$\Pr_{x \sim \mathcal{D}}[t(x) = d_t[i + 1 :] \circ \gamma_i \circ d_h[i] \circ \alpha_{i-1} \circ d_t[: i](x)] \geq 1 - \epsilon$$

We propose two additional axioms, namely, Axioms 5 and 6, that are more informal in nature and we do not consider

them for the analysis presented in this paper. These axioms are presented as a goal for future work and described in Appendix B.

**Checking the Axioms.** The proposed axioms can be checked statistically. Given a test dataset (which need not be labeled), we can estimate the error rate $\epsilon$ for each axiom by computing the proportion of test inputs which violate the equality condition specified by the axiom. The number of violations is distributed binomially—the parameter $p$ of this binomial distribution is equal to $\epsilon$. Hence, any method for deriving confidence intervals for the parameter of a binomial distribution can be used to derive confidence intervals for our estimate of $\epsilon$; we use the Clopper-Pearson method (Clopper & Pearson, 1934) in our experiments. Validating the axioms is thus cheap and feasible in practice. These axioms represent a minimal set that we believe are necessary albeit may not be sufficient in all cases for characterizing a mechanistic interpretation. In future work, we plan to address this by considering analysis of models trained for other problems.

**Relationship to Existing Approaches.** The evaluation of Nanda et al. (2023a)'s seminal paper on mechanistically interpreting the complete behavior exemplifies typical approaches to evaluating a mechanistic interpretation (Zhong et al., 2023; Quirke & Barez, 2024; Chughtai et al., 2023). The first three pieces of evidence (presented in Sections 4.1, 4.2, and 4.3 in their paper) resemble our Axiom 2 that compares each individual component of the mechanistic interpretation with the corresponding component of the model. However, as there does not exist a direct analogue of abstraction, this analysis is primarily observational.

The evidence they present in Section 4.4 of their paper is similar to our Axiom 4 that checks the effect on the output of the model when individual model components are replaced by the corresponding components from the mechanistic interpretation. Notably, they do not present the evidence required by our Axioms 1 and 3. This amounts to not considering the compositional structure of the model and, therefore, the possible compounding of error introduced by each component of the mechanistic interpretation; we discuss this further at the end of this section.

Causal abstraction (Geiger et al., 2021; 2022; 2024; 2025) and causal scrubbing (Chan et al., 2022) are key existing attempts to formalize the notion of a mechanistic interpretation. Both of these formulate the abstract and concrete models as directed acyclic graphs (DAG) with a map that relates the two DAGs. While it may seem that, compared to our approach, the DAG formulation is more generic since it admits parallel composition of components in addition to sequential composition, this is not fundamental, in particular, Appendix B.1 discusses how to represent arbitrary computational graphs in our framework.

The map used by causal abstraction to relate the two DAGs corresponds to our abstraction function $\alpha$; however, it does not have any analog of a concretization function $\gamma$. Geiger et al. (2025) note that the inverse of abstraction may be used as a concretization operator; however, this may not be feasible in practice, as this choice necessitates the use of set semantics for evaluation of the concrete model when the abstraction operator fails to be invertible. Under the causal abstraction framework, a valid interpretation is defined by comparing the effects of interventions on the concrete model with corresponding interventions on the abstract model; in contrast, our notion of a valid interpretation may be viewed as one which is invariant up to a class of interventions, characterized by our axioms, which interleave the concrete and abstract models. While causal abstraction is a very general framework for the validation of mechanistic interpretations, in practice, it uses the specific metric of interchange intervention accuracy (Geiger et al., 2022) to validate interpretations and this metric fails to directly evaluate the equivalence of internal representations.

On the other hand, the map defined by causal scrubbing can be seen as an form of concretization, but it does not define an abstraction function. While both these approaches compare the outputs of the concrete and abstract models similar to our Axioms 3 and 4, they tend to omit direct evaluation of the equivalence of internal representations, as in our Axioms 1 and 2. This is potentially problematic since there is no direct validation that the intermediate steps of the mechanistic interpretation correspond to the intermediate steps of the neural network (Scheurer et al., 2023). Further, as causal abstraction and scrubbing lack concretization and abstraction operators respectively, even Axioms 3 and 4 cannot be expressed directly in these frameworks.

**Compositionality.** While it may appear that our axioms for compositional evaluation (Axioms 1 and 3) follow from their componentwise counterparts (Axioms 2 and 4 respectively), this is not the case; for example, while bounds of the form of Axiom 1 can be derived from Axiom 2, these hold only with far weaker $\epsilon$; when the number of components is sufficiently high, the derived bounds entirely cease to be meaningful. More details are given in Appendix E; Appendix F includes empirical evidence demonstrating the pitfalls of omitting the compositional axioms.

## 4. Case Study: 2-SAT

### 4.1. Data and Model Details

**The 2-SAT Problem.** Given a Boolean formula, i.e., a conjunction of disjunctive clauses over exactly two literals, the 2-SAT problem is to determine whether there is an assignment to the formula's variables that makes the formula evaluate to True; the formula is said to be satisfiable, or SAT.

For example, the formula $(x_0 \vee x_1) \wedge (x_1 \vee \neg x_2)$ is satisfiable, with a satisfying assignment $x_0 = $ True, $x_1 = $ False, $x_2 = $ False (written also as $x_0, \neg x_1, \neg x_2$). If a formula has no satisfying assignment, it is said to be unsatisfiable, or UNSAT.

**Dataset.** We construct a dataset of randomly generated 2-SAT formulas with ten clauses and up to five variables, eliminating syntactic duplicates. We use a solver (Z3 (De Moura & Bjørner, 2008)) to check satisfiability for each formula. We built a balanced dataset of $10^6$ SAT and $10^6$ UNSAT instances; we used 60% (also balanced) for training, and the rest for testing. For model analysis, we construct a separate dataset with $10^5$ SAT and $10^5$ UNSAT instances split as before in to train and test sets.

Each formula is represented as a string. For example, $(x_0 \vee x_1) \wedge (x_1 \vee \neg x_2)$ is represented as "$(x_0 x_1)(x_1 \neg x_2) : s$" where $s$ indicates that the formula is SAT ($u$ would indicate UNSAT). We tokenize these formulas by considering each $x_i$, its negation $\neg x_i$, and each of the symbols in the set $\{(,), :, s, u\}$ as a separate token. The colon token ('$:$') is the final token in the input and we consider the next token predicted by the model as the output; hence, we refer to this token as the *readout* token and refer to its position as the readout position.

**Model Architecture and Training.** We use a two-layer ReLU decoder-only Transformer with token embeddings of dimension $d = 128$, learned positional embeddings, one attention head of dimension $d$ in the first layer, four attention heads of dimension $d/4 = 32$ in the second, and $n = 512$ hidden units in the MLP (multi-layer perceptron). We use full-batch gradient descent with the AdamW optimizer, setting the learning rate $\gamma = 0.001$ and weight decay parameter $\lambda = 1$. We perform extensive epochs of training (1000 epochs), recognizing the combinatorial nature of SAT problems and the need for thorough exploration of the solution space. We obtained a model with 99.76% accuracy on test data. All our experiments were run on an NVIDIA A100 GPU.

### 4.2. Mechanistic Interpretability Analysis

The network can be naturally decomposed into its blocks and we describe the analysis of each in the following two sections. We note that our axioms help guide our analysis, particularly for the second block. As our goal is to understand the algorithm used by the model to determine satisfiability, we refrain from analyzing behavior which does not affect the final prediction. Thus, we focus on the model's prediction at the readout position. Figure 5 in Appendix D describes our decomposition of the model into the concrete components $d_t[i]$.

#### 4.2.1. FIRST BLOCK AS PARSER

To reverse-engineer the first block's behavior, we start by examining the attention patterns, i.e., the attention scores (by default, post-softmax unless stated otherwise) calculated by both blocks on given input formulas. Figure 1 shows the attention scores on a formula from our test set. We clearly see patterns emerge—most tokens in the first block attend heavily to the first token, while tokens in position $4i + 2$, $0 \le i < 10$, that we refer to as the *second literal positions*, primarily attend to themselves and the previous token, i.e., to the two literals in each clause.[3] Attention from the readout token is nearly uniform. This suggests that the first block parses each clause, storing the parsed clause information in the token representations output by the first block in the second literal positions.

In the second block, the attentions heavily focus on these second literal positions, suggesting that the model uses the parsed clause information contained in these representations to classify. This allows us to restrict our analysis of the first block to the second literal positions and the readout position. To further validate this decision, we compute the average attention score from each head on each token across the test set (see Figure 6 in Appendix D.1). The results show strong periodicity with period 4, following the pattern seen in Figure 1.

We perform additional distributional and worst-case analysis of attention patterns (described in Appendix D.1) and these analyses confirm the parsing hypothesis for the first block.

**Interpretation.** Based on our analysis of the attention patterns for the first block, we hypothesize that it is parsing the input sequence of tokens into a list of clauses (Listing 5 in Appendix D.1). The behavior of attention is enough to form this hypothesis, as the sparse attention patterns show that information primarily flows from tokens in a clause to the token in the second literal position in the same clause; hence we can view the action of the first block as computing a representation for each clause in the input. In this sense, analyzing the MLP in the first block is not necessary to identify the semantic function of the first block.

We show in Sec 4.2.3 that we can successfully validate this hypothesis using our axioms. As we'll see in the analysis of the second block in Sec 4.2.2, we cannot always count on self-explanatory attention patterns; for the second block, we will be forced to analyze the much harder-to-interpret MLP.

#### 4.2.2. SECOND BLOCK AS EVALUATOR

From the first block's analysis, we know that the inputs to the second block can be described, in an abstract sense, as

---

[3]Formulas are of the form, $(x_0 x_1)(x_1 \neg x_2) : s$, so token positions $4i + 1$ and $4i + 2$ correspond to the literals of the $i^{\text{th}}$ clause.

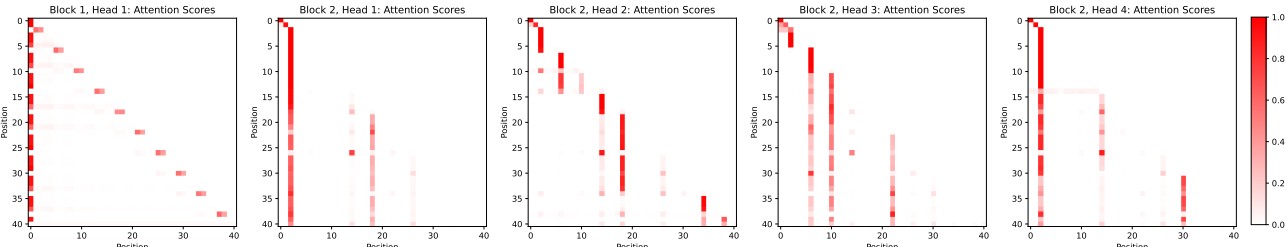

Figure 1: Attention scores for all heads on the first sample of the test set, the formula $(\neg x_0 \neg x_1)(x_1 \neg x_4)(x_1 x_2)(x_0 x_3)(\neg x_2 \neg x_3)(x_2 \neg x_4)(\neg x_0 \neg x_3)(x_0 x_2)(x_1 \neg x_2)(x_1 x_4)$. The x-axis represents the source (key) positions and the y-axis represents the destination (query) positions for the attention mechanism.

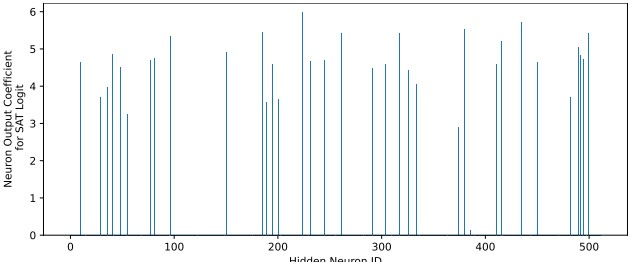

Figure 2: Output coefficients of hidden neurons computed via the composition of the output layer of the MLP and the unembedding matrix projected to the SAT logit.

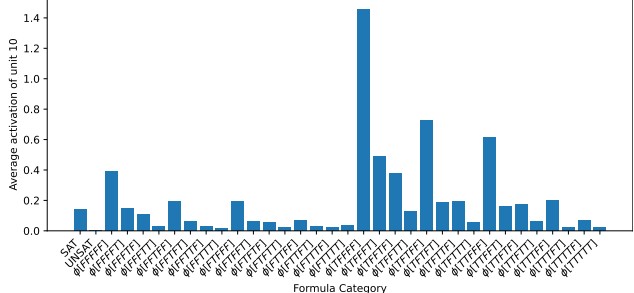

Figure 3: Average activation of neuron 10 from SAT formulas, UNSAT formulas, and formulas satisfiable with particular assignments to the variables.

the list of clauses in the formula. As the model's output is a label SAT or UNSAT, the interpretation of the second block is a function that checks satisfiability given a list of clauses. This means that the core logic of the model is encoded primarily in the second block.

**Identifying the Key Pathway.** We observe that the unembedding vector for UNSAT is nearly the negative of that of SAT (in particular, $\|W_U^{\text{SAT}} + W_U^{\text{UNSAT}}\| < 10^{-5}$ and their cosine similarity is -1 up to floating point precision), and hence, their logits are likewise negatives of each other. Also, for all formulas in the dataset, either the SAT or the UNSAT logit output at the readout position is much larger than the logits for all other tokens. Hence, the effect of components on classification can be fully described by their effects on the SAT logit. Furthermore, we observe that the effect of the attention mechanism on the SAT logit is consistently suppressive and varies little between samples. Hence, the core pathway necessary to identify SAT flows through the MLP at the readout position. Unlike the first block, we cannot identify an interpretation directly from the behavior of attention. See Appendix D.2 for more details.

**Sparsity in the MLP Hidden Layer.** We begin our analysis of the MLP by noting that, somewhat surprisingly, it exhibits sparsity in the hidden neurons. We conclude this by first observing that the second layer of the MLP and the unembed-

ding operation are both linear, so their composition is again linear. From this perspective, then, the MLP's effect on the classification is fully captured by a weighted sum of the post-activation outputs of the hidden neurons. We refer to these weights as *neuron output coefficients* for the SAT logit. This is the sense in which we observe sparsity; to see this clearly, see Figure 2. There are only 35 neurons which have a non-negligible absolute coefficient ($> 10^{-6}$); one of these has a significantly lower coefficient than the others along with consistently low activation values on the samples in our analysis training set resulting in a negligible effect on the SAT logit. Hence, we restrict our analysis to the remaining 34 neurons.

**Evaluation in the Hidden Neurons.** Note furthermore that all the non-negligible coefficients are positive; hence, the effect of each of the relevant neurons is *purely* to promote SAT. This suggests that the neurons may each recognize some subset of SAT formulas instead of recognizing characteristics of UNSAT formulas. Figure 3 shows that the typical activation of one such neuron on SAT formulas is highly dependent on the assignment to the variables which satisfies the formula. This suggests that the MLP may implement a very simple algorithm: that of exhaustive evaluation. In particular, if we treat the formula $\phi$ as a Boolean function $f_\phi$, $\phi$ is SAT if and only if $f_\phi$ is not the constant False function. Hence, we can identify SAT formulas by a brute-force technique which computes the truth table of the corresponding

Boolean functions and checks whether any entry is True.

We'll see that this is indeed the case; hence, we will refer to these neurons as *evaluating neurons*. However, the behavior of the neurons is somewhat more complex than the most natural form, in which each neuron specializes in identifying formulas satisfiable with one of the 32 possible assignments. In particular, as seen in Figure 3, some neurons are affected by multiple assignments to the variables. The notation $\phi[...]$ means that the corresponding Boolean function $f_\phi$ evaluates to True when the variables are assigned the shown truth values. For instance, $\phi[TTFFT]$ indicates that formula $\phi$ is satisfiable with $x_0, x_1, x_4$ set to True and $x_2, x_3$ set to False.

A natural hypothesis is then that each of the evaluating neurons evaluates $\phi$ on some set of assignments, with some overlap between neurons; we refer to such interpretations of neuron activation behavior as *disjunction-only* since they take the form $h ::= \phi[...] \mid h \vee h$. However, we observe that the actual behavior is less intuitive: in some cases, satisfiability with an assignment can *decrease* activation on some evaluating neurons. Hence, we also consider more general Boolean expressions in terms of the $\phi[...]$'s as neuron interpretations. *We find that our axiomatic analysis provides a clear way to quantify the trade-off between interpretability and fidelity of the different interpretations.*

As decision trees can represent arbitrary Boolean functions, we use standard decision tree learning to learn the more general interpretations of neurons; specifically, we learn classifiers trained to predict whether a neuron's activation is above or below a threshold (0.5 in our experiments) given all the features $\phi[...]$ for a formula $\phi$. Deriving the simpler disjunction-only interpretation is easier; we do so by identifying the assignments $a$ such that the empirical mean, calculated over our analysis training set, of a neuron's post-activation value on formulas satisfied by $a$ is above the threshold. For instance, in Figure 3, these are the assignments that cause the post-activation score to be above the threshold.

Finally, we observe that the output coefficients of the evaluating neurons for the SAT logit are consistently high (> 2.9 in all cases), which means that a high activation on any individual evaluating neuron forces prediction of SAT; see Appendix D.2 for more details. In this sense, the action of MLP's output layer and the unembedding layer is conceptually close to logical OR operation. We now have enough information to describe our interpretation of the second block.

**Interpretation.** Based on our analysis, we decompose the interpretation of the second block into two components. We hypothesize that the second block up to the MLP's hidden layer evaluates the input formula with different assignments and outputs the results of these evaluations as a list of Booleans. The neuron interpretations describe the precise combination of assignments considered for computing each Boolean output (see Tables 2 and 3 of Appendix G for two different neuron interpretations). We evaluated the completeness of the neuron interpretations, i.e., their ability to correctly evaluate all 2-SAT formulas, using Z3 (De Moura & Bjørner, 2008). The rest of the model, namely, the MLP's output layer and the unembedding layer, performs a logical OR over the Booleans output by the previous component and outputs True if the formula is SAT. Listing 6 and 7 in Appendix D.2 describe the mechanistic interpretation of the second block and the entire model, respectively, using Python syntax.

### 4.2.3. CHECKING THE AXIOMS

**First Block.** Since the input to the neural model is a sequence of token IDs, $\alpha_0$ here is the function that translates token IDs into tokens. To derive $\alpha_1$ and $\gamma_1$, we first compute a canonical representation for each possible clause $\psi$.

Drawing from the observation that the output of attention on the second token of each clause is largely a function of the two tokens in the clause alone (ignoring all other past tokens), we compute the representation output by block 1 for clause $\psi$ in each position $i$ by masking attention such that it is only applied to the left and right literals in the clause. The canonical representation for $\psi$ is the average of the representations derived for each position $i$. $\alpha_1 : [\texttt{Tensor}] \rightarrow [(\texttt{Literal}, \texttt{Literal})]$, where $\texttt{Literal}$ is the type consisting of the $x_i$ and their negations, maps the list of representations output by the first block to a list of clauses by comparing (via cosine similarity) the representations in the second-variable positions $4i + 2$ with the canonical clause representations and returning the clauses with the most-similar representations. $\gamma_1 : [(\texttt{Literal}, \texttt{Literal})] \rightarrow [\texttt{Tensor}]$ maps a list of clauses to a list of canonical clause representations. For positions other than $4i + 2$, $\gamma_1$ returns the mean representation in that position across the training set. For position $4i + 2$, it returns the canonical representation for the clause in position $i$. See Listing 1 and Listing 3 of Appendix D for pseudocode for $\alpha_1$ and $\gamma_1$, respectively.

**Prefix Equivalence.** For each clause position $i$, $\alpha_1$ outputs a clause $\phi := l \vee r$, where $l$ and $r$ are literals, and, as such, we consider the clauses $r \vee l$ and $l \vee r$ equal. With this notion of equality, we obtain an $\epsilon^4$ of approximately 0.0000374 (corresponding to perfect matching on the test set) between the abstract states output by $\alpha_1 \circ d_t[1]$ and $d_h[1] \circ \alpha_0$. When we enforce order consistency, we obtain a much worse $\epsilon$ of 0.849; however, as the abstract state is order independent and the second block's behavior does not vary significantly with the order of variables in a clause, an order-dependent notion of equality is not necessary here.

---

[4]We always report $\epsilon$ in terms of the upper bound of a one-sided 95% Clopper-Pearson confidence interval.

**Component Equivalence.** As this is the first component, component equivalence and prefix equivalence are identical, hence the results are the same as above.

**Prefix Replaceability.** We obtain $\epsilon \approx 0.0418$, i.e. substituting the first component of the model with $\gamma_1 \circ d_h[1] \circ \alpha_0$ affects the final output 4.2% of the time. In particular, this quantifies the effect of replacing the actual first-block representation of ':' with its mean value across the dataset and the effect of ignoring attention to previous clauses when computing the canonical clause representations.

**Component Replaceability.** As this is the first component, component and prefix replaceability are identical, hence the results are the same as above.

**Hidden Layer of Second Block.** The output of the second concrete component is a pair consisting of the residual from attention and the post-activation outputs of the hidden neurons; $\alpha_2 : (\texttt{Tensor}, \texttt{Tensor}) \to [\texttt{bool}]$ maps these to a Boolean vector representing whether or not sufficiently high activation ($> 0.5$) has occurred at each of the 34 evaluating neurons. $\gamma_2 : [\texttt{bool}] \to (\texttt{Tensor}, \texttt{Tensor})$ simply returns a constant value for the residual (the mean on the analysis training set); the MLP's hidden neurons have zero activation except for the evaluating neurons with True values in the corresponding position in the abstract state vector, which we assign an arbitrary large activation (2 in our experiments). Note that this is higher than the threshold for $\alpha_2$; we observe that this amplification is necessary for a reliable interpretation; see below. See Listing 2 and Listing 4 of Appendix D for pseudocode for $\alpha_2$ and $\gamma_2$, respectively.

**Prefix Equivalence.** We obtain $\epsilon \approx 0.182$ when using the decision tree neuron interpretations, compared with approximately 0.309 for the disjunction-only interpretations in which our neuron models activate on formulas satisfiable with any of a set of assignments. This means that the disjunction-only interpretations are much worse at predicting the intermediate state of the model. This is the only place where this simplifying choice has a cost with respect to the validity of the mechanistic interpretation; aside from component equivalence below, the disjunction-only interpretations differ little from the decision trees in all other respects. This illustrates the need to check the axiomatic properties at component level for the model under analysis.

**Component Equivalence.** We again obtain $\epsilon \approx 0.182$ for the decision tree interpretations and 0.309 for the disjunction-only interpretations. The similarity in the $\epsilon$ values for prefix and component equivalence is because the interpretation of the first component perfectly matches the behavior of neural network, up to literal order, on the analysis test set.

**Prefix Replaceability.** In both cases, substituting the first two concrete components with the first two abstract

components has minimal effect on the model's output behavior: specifically, we get $\epsilon \approx 0.0128$ for the decision tree interpretations, and $\epsilon \approx 0.00290$ for the disjunction-only interpretations. If we drop the amplification step discussed above for $\gamma_2$, we significantly affect the model's predictions: we get an $\epsilon$ of approximately 0.249 (i.e. over 24% of samples are affected) for the decision tree interpretations and approximately 0.135 for the disjunction-only interpretations.

**Component Replaceability.** As above, we obtain $\epsilon \approx 0.0128$ for the decision tree interpretations, and $\epsilon \approx 0.00290$ for the disjunction-only interpretations.

**Output Layer of Second Block.** For our analysis, we consider the model to be composed with a function which returns True if and only if the top logit at the readout position is SAT. The output type of this composed model and the mechanistic interpretation are identical; hence $\alpha_3$ and $\gamma_3$ are both the identity function.

**Prefix Equivalence.** Since prefix equivalence incorporates the mechanistic interpretation of the previous component, we need to consider both the decision tree and the disjunction-only version of the previous component. We obtain very close matching in both cases, with $\epsilon \approx 0.0128$ for the decision tree interpretations and $\approx 0.00290$ for the disjunction-only interpretations.

**Component Equivalence.** We obtain a very close match with $\epsilon \approx 0.00433$.

**Prefix Replaceability.** As this is the final component of the network and $\alpha_3$ and $\gamma_3$ are the identity, this is the same as prefix equivalence.

**Component Replaceability.** Same as component equivalence for the reason above.

## 5. Case Study: Modular Addition

Nanda et al. (2023a) train a model with a single Transformer block to perform modular addition. The model is given input of the form $a\ b\ =$ and returns $a + b \mod P$ at the '=' token where $P$ is fixed to be 113. They claim the following mechanistic interpretation of the concrete model (see Listing 8 in Appendix H for the pseudocode for the abstract components):

1. **Encoding of inputs**: The embedding matrix represents $a$ and $b$ as $\sin(w_k a)$, $\sin(w_k b)$, $\cos(w_k a)$, and $\cos(w_k b)$ for key frequencies $w_k = (2k\pi)/P$ for $k \in \{14, 35, 41, 42, 52\}$.
2. **Computation of sum-of-angles identities**: The attention and MLP input layer compute $\cos(w_k(a+b))$ and $\sin(w_k(a+b))$ using trigonometric identities.
3. **Difference-of-angles identities and argmax**: The MLP output layer and the unembedding matrix

computes $\cos(w_k(a + b - c))$ for each key frequency and for each $c \in \mathbb{Z}_P$, adds the computed cosines, and then selects the $c$ maximizing the sum, which occurs at $c^* = a + b \mod P$.

As all abstract components but the last one output continuous values, we must discretize the output of each abstract component for Axioms 1 and 2 to be meaningful. For the first component, we apply a simple discretization of rounding up to three decimal points; for the second component, we define an equivalence class which treats abstract states as equivalent when their downstream effects on both the abstract and concrete models are identical. Discretizations of this form are quite general: an equivalence class up to the next discrete-valued intermediate value is expressible for any abstract model with a discrete-valued output. Axioms 3 and 4 ensure that any discretization does not impact end-to-end behavior. The $\alpha$ and $\gamma$ functions for each component are linear maps learned from data. See Appendix H for more details.

We apply our approach to this model and confirm that the Nanda et al. (2023a)'s mechanistic interpretation satisfies our axioms, noting that, in their paper, while Nanda et al. (2023a) provide evidence similar to our Axioms 2 and 4 in support of the validity of their interpretation, they do not provide any evidence as required by our Axioms 1 and 3.

In particular, with the above discretization we obtain a very strong $\epsilon$ of 0.000335, which corresponds to perfect matching. However, it is important to note that results with simple discretizations indicate that the model's behavior is not fully captured. In particular, we obtain an $\epsilon$ of 1 from Axioms 1 and 2 for the second component when, instead, we discretize by rounding to a single decimal place. There are important tradeoffs to the application of this type of discretization: while it enables the analyst to eliminate variation which has no impact on the downstream behavior of the model, it also limits the ability of Axioms 1 and 2 to validate the internal behavior.

## 6. Conclusion

We presented a set of axioms that characterize a mechanistic interpretation and enable judgment of the interpretation's validity with respect to the neural network under analysis. Using the axioms as a guide, we analyzed a Transformer-based model trained to solve the 2-SAT problem. We applied our axioms to validate the mechanistic interpretation of this 2-SAT model and a model trained to solve the modular addition task. Our axioms provide an automated and quantitative way of evaluating the quality of a mechanistic interpretation (via the $\epsilon$ values). Not only can the $\epsilon$'s serve as useful progress measures (in the sense of Nanda et al. (2023a)) to understand the training

dynamics of models but can also help in the development of techniques for automated mechanistic interpretability analyses by serving as a useful evaluation metric.

## Impact Statement

We believe that work on mechanistic interpretability in particular and on internal interpretability of neural networks in general is essential for increasing trust in neural networks and mitigating their risks. In this context, the work presented in this paper is of a foundational nature and advances the interpretability research agenda by clarifying the notion of a mechanistic interpretation. Therefore, we do not foresee a direct path from our work to negative societal impacts.

## Acknowledgments

Nils Palumbo and Somesh Jha are partially supported by DARPA under agreement number 885000, NSF CCF-FMiTF-1836978 and ONR N00014-21-1-2492.

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

# A. More Related Work

**Probing.** Probing involves training a separate supervised classifier to predict human-understandable features of the data from the model's internal activations (Alain & Bengio, 2017; Zou et al., 2023) and has found wide applications for understanding the representations learned by language models (Tenney et al., 2019; Li et al., 2023; Nanda et al., 2023b; Burns et al., 2023; Belrose et al., 2023; Huben et al., 2024; Gurnee et al., 2023; Wang et al., 2024). Even though a probe might be successful in predicting features from internal activations, it need not imply a causal relationship between these features and the model output (Belinkov, 2022). It has also been questioned whether the model activations genuinely represent the feature of interest or the probe itself learns these features by picking up on spurious correlations (Hewitt & Liang, 2019). However, recent works have demonstrated that for Transformer-based language models probes can be used to steer the model's output behavior by manipulating the identified internal representations (Zou et al., 2023; Li et al., 2023).

Related to probing is the work on *concept-based reasoning* (Kim et al., 2018; Crabbé & van der Schaar, 2022; Yeh et al., 2021; 2020) that extracts representations of high-level concepts as geometric shapes in the neural network representation space (i.e., the activation spaces of inner layers of neural networks) and quantifies the impact of these high-level concepts on model outputs via attribution-based techniques.

**Attention Pattern Analysis.** There is a large body of work that attempts to understand the behavior of Transformer-based models by analyzing the *attention patterns* computed by the self-attention layer (Allen-Zhu & Li, 2023; Zhang et al., 2023; Ebrahimi et al., 2020; Vig & Belinkov, 2019; Hao et al., 2021; Qiang et al., 2022; Lu et al., 2021; Abnar & Zuidema, 2020; Chefer et al., 2021; Liu et al., 2023). However, there is an active ongoing debate about the validity of attention patterns as a tool for interpreting model behavior (Jain & Wallace, 2019; Pruthi et al., 2020; Bibal et al., 2022; Bastings & Filippova, 2020; Wiegreffe & Pinter, 2019).

# B. Axioms of Mechanistic Interpretation

Axioms 5 and 6 are stated more informally. Unlike the earlier axioms requiring the observed behaviors of the mechanistic interpretation and the neural network to coincide, these axioms are concerned with the *compilability* of the mechanistic interpretation into a neural network. Both axioms assume the existence of a semantics-preserving compiler from $\lambda_H$ to $\lambda_T$ (denoted by $compiler_{\lambda_T}^{\lambda_H}$). While Axiom 5 requires the compiled version of $h$ to have the same structure and parameters as $t$ (i.e., syntactic equivalence), Axiom 6 requires the compiled version to only have the same structure as $t$. The requirements imposed by these last two axioms are hard to establish in practice, and we do not consider them for the analysis presented in this paper. We present these axioms as a goal for future work. Recent work on compilers from the RASP language (Weiss et al., 2021) to Transformer models is a promising step in this direction (Lindner et al., 2023).

**Axiom 5** (Strong Mechanistic Derivability). $h$ is mechanistically derived from $t$ if there exists a semantics-preserving compilation $compiler_{\lambda_T}^{\lambda_H}(h)$ of $h$ in $\lambda_T$ such that $compiler_{\lambda_T}^{\lambda_H}(h)$ and $t$ are syntactically equal.

**Axiom 6** (Weak Mechanistic Derivability). $h$ is mechanistically derivable from $t$ if there exists a semantics-preserving compilation $compiler_{\lambda_T}^{\lambda_H}(h)$ of $h$ in $\lambda_T$ such that $compiler_{\lambda_T}^{\lambda_H}(h)$ has the same *architecture* (with the same number of parameters) as $t$.

## B.1. Extension to Circuits

While we focus on end-to-end analyses, our axioms are compatible with the analysis of subgraphs of a larger model.

### B.1.1. AXIOMS FOR ARBITRARY COMPUTATIONAL GRAPHS

We can extend our axioms to operate directly on the graphs $G$ and $G'$; we present a generalization of Axioms 1 through 4 below. Let $G = (V, E)$ be the concrete model expressed as a computational graph and let $G' = (V', E')$ be the computational graph, isomorphic to $G$, representing the abstract model. Let the isomorphism between $G$ and $G'$ be $\Pi$ and call the input and output vertices of $G$, $in$ and $out$, respectively; while this can be easily extended to multiple input and output vertices, we present the single input, single output case for simplicity. Let $execute(G)(x)$ return a mapping $val : V \rightarrow Val$ with the values of all intermediate nodes where $Val$ is the set of all possible values (i.e., vectors of reals) that the nodes can take. For a (partial) assignment to the vertices $assign : V' \rightarrow Val$ with $\{in\} \subseteq V' \subseteq V$, $propagate(G)(assign)$ returns a mapping of the same type $val : V \rightarrow Val$ which is the result of execution of $G$ where the provided intermediate assignments override the standard results of execution for those vertices. In this way, we can represent arbitrary interventions on $G$ and $G'$.

We define an abstraction operator $\alpha_v$ for each concrete node $v \in V$ and a concretization operator $\gamma'_v$ for each abstract node $v' \in V'$. $\alpha$ and $\gamma$ are the corresponding functions which apply the corresponding mappings to each vertex in the graph. Finally, define $select(v)(val) = val(v)$ for $val : V \to Val$ and $v \in V$; we overload and let $select(V')(val)(v') = val(v')$ for $V' \subseteq V$ and $v' \in V'$.

We can then define extensions of Axioms 1 to 4 below:

**Axiom 7** ($\epsilon$-Prefix Equivalence). $\forall v \in V$.

$$\Pr_{x \sim \mathcal{D}}[\alpha_v \circ select(v) \circ execute(G)(x) = select(\Pi(v)) \circ execute(G') \circ \alpha_{in}(x)] \geq 1 - \epsilon$$

**Axiom 8** ($\epsilon$-Component Equivalence). $\forall v \in V$.

$$\Pr_{x \sim \mathcal{D}} \left[ \begin{array}{l} \alpha_v \circ select(v) \circ execute(G)(x) = \\ select(\Pi(v)) \circ propagate(G') \circ select(\Pi(p) \cup \{\Pi(in)\}) \circ \alpha \circ execute(G)(x) \end{array} \right] \geq 1 - \epsilon$$

where $p = predecessors(v)$.

**Axiom 9** ($\epsilon$-Prefix Replaceability). $\forall v \in V$.

$$\Pr_{x \sim \mathcal{D}} \left[ \begin{array}{l} select(out) \circ execute(G)(x) = \\ select(out) \circ propagate(G) \circ select(\{in, v\}) \circ \gamma \circ execute(G') \circ \alpha_{in}(x) \end{array} \right] \geq 1 - \epsilon$$

**Axiom 10** ($\epsilon$-Component Replaceability). $\forall v \in V$.

$$\Pr_{x \sim \mathcal{D}} \left[ select(out) \circ execute(G)(x) = \left( \begin{array}{l} select(out) \circ propagate(G) \circ select(\{in, v\}) \circ \\ \gamma \circ propagate(G') \circ select(\Pi(p) \cup \{\Pi(in)\}) \circ \\ \alpha \circ execute(G)(x) \end{array} \right) \right] \geq 1 - \epsilon$$

where $p = predecessors(v)$.

Note that our Axioms 1 to 4 are equivalent to these formulations, specialized to linear computational graphs.

In linear computational graphs, all dependencies on ancestor nodes are mediated by a node's predecessor; hence, there is a limited need to consider *parallel interventions*, which involve simultaneously performing independent interventions on multiple nodes of the graph. In the case of prefix equivalence and prefix replaceability, any parallel interventions on linear graphs are equivalent to the intervention on the final node in the sequence and hence have no additional effect. While parallel extensions of component equivalence and component replaceability would strengthen evaluation in the linear setting, considering parallel interventions is particularly important in the nonlinear case.

Consider the following scenario: there are two sibling nodes in the graph that encode redundant computations and the concrete states output by both these nodes contain information needed by the downstream circuit. Also, let us say that the corresponding abstract model fails to faithfully capture the computation performed by either of these nodes. However, this failure may not be captured by the axioms without parallel interventions since the redundant copy of the computation in the sibling concrete node masks any intervention applied to the other concrete node.

One potential solution is to merge such redundant sibling nodes in the analyzed graph $G$, however, this prevents checking equivalence of the nodes independently. A more general solution is to extend Axioms 7 through 10 to accommodate arbitrary parallel interventions. We present a preliminary formulation of an axiom for parallel intervention below; a final formulation is left to future work. In particular, this axiom cannot be efficiently evaluated in its current form.

We first extend *execute* and *propagate* to accommodate execution of arbitrary mixtures of the graphs $G$ and $G'$. These behave as before, except that any concrete inputs to an abstract operation are abstracted prior to execution, and, similarly, any abstract inputs to a concrete operation are concretized prior to evaluation. Let $interleave(G, G', V')$ be a graph identical to $G$ except that the operations for any nodes $v \in V'$ are replaced with the operations for the corresponding nodes $\Pi(v)$ in $G'$. Let $conditional\_abstract(V')$ be a mapping on the values of the nodes $v \in V$ identical to $\alpha_{\Pi(v)}$ for $v \notin V'$ and the identity if $v \in V'$. We can now express an axiom which generalizes Axioms 7, 8, and 10 to parallel interventions:

**Axiom 11** ($\epsilon$-Equivalence). $\forall V' \subseteq V, \forall v \in V$.

$$\Pr_{x \sim \mathcal{D}} \left[ \begin{array}{l} select(v) \circ conditional\_abstract(V') \circ execute(G)(x) = \\ select(v) \circ conditional\_abstract(V') \circ execute(interleave(G, G', V'))(x) \end{array} \right] \geq 1 - \epsilon$$

The condition of Axiom 11 can be made even stronger by the treeification (Chan et al., 2022) of the graphs $G$ and $G'$ prior to evaluation. This allows, for each node $v \in V$, independent selection of the set of ancestors whose operations are replaced with their abstract equivalents. With treeification, Axiom 11 can express prefix replaceability as well, leading it to generalize Axioms 7 through 10.

We can directly validate arbitrary circuits with the standard linear-graph axioms. In particular, we will describe how, for any computational graphs $G = (V, E)$ and $G' = (V', E')$, which represent the concrete model $t$ and the abstract model $h$, respectively, we can construct a linearized computational graph. It is enough to show the construction for $G$.

We will do so by propagating partial assignments to the vertices $v \in V$ through a topologically sorted sequence of operations which compute the values of each node $v$ as a function of the values of the predecessors. First, we extend $\lambda_T$ to include multivariate functions $f(x_1, x_2, \ldots, x_n)$ for $x_i \in Var$. Suppose that $op : V \to \lambda_T$ returns the operation $op(v)$ computed by node each node $v$, and let $predecessors(v)$ return the predecessors of node $v$ in the argument order of $op(v)$. For simplicity, we assume that $G$ has exactly one input node $in$ and exactly one output node $out$.

Let $o : [|V|] \to V$ be a topological ordering of $G$. Let $idx : V \to [|V|]$, the inverse of $o$, return the position of each node $v$ in the topological ordering $o$. We again extend $\lambda_T$ to include the function $execute\_op(f, p, v)(env)$ which, given an $m$-ary function $f$, a node $v \in V$, a partial assignment to the vertices $env : V \to Val \cup \{\bot\}$ and a sequence of input nodes $p : [m] \to V$ such that $env(p(i)) \in Val$ for each $1 \le i \le m$, returns an updated environment $env'$ of the same type, identical to $env$ except that $env'(v) = f(env(p(1)), \ldots, env(p(m)))$. Finally, for $val \in Val$, let $input\_env(val)$ be a partial assignment to the vertices $input\_env(val) : V \to Val \cup \{\bot\}$ which is $\bot$ everywhere but $in$, where we have $input\_env(val)(in) = val$.

Noting that $op(1)$ must always be $in$, we can define a decomposition of $t$ as follows:

$$d_t = (select(out) \circ execute\_op(op(o(|V|)), predecessors(o(|V|)), o(|V|)))$$
$$\circ \ldots$$
$$\circ execute\_op(op(o(2)), predecessors(o(2)), o(2))$$
$$\circ input\_env.$$

## C. Remark on Extensional Equivalence

Scheurer et al. (2023) presents an example illustrating the *extensional equivalence* problem: in particular, they demonstrate that causal scrubbing cannot distinguish between two mechanistic interpretations with equivalent input-output behavior but which differ in internal behavior. See Figure 4 for the two models. We make copies of the $x_0$ and $x_1$ nodes in model (b) to ensure that the two models are isomorphic.

As observed by Scheurer et al. (2023), the two are indistinguishable to causal scrubbing, noting that we assume that $x_0$ and $x_1$ are both positive. This is a necessary consequence of the resampling technique used by causal scrubbing to derive its interventions. In particular, causal scrubbing replaces the values of concrete nodes with randomly selected values which are equivalent up to the abstract model.

Clearly, the concrete model is invariant with respect to resampling ablations derived from interpretation (a), which is identical to the concrete model. But, likewise, there is no effect from the corresponding intervention derived from interpretation (b). For instance, resampling concrete values of node (4), $\frac{1}{x_0}$, up to equivalence in the abstract state of $x_0$ can have no effect. Hence, both interpretations are evaluated as perfect by causal scrubbing. For the same reason, these interpretations cannot be distinguished by interchange intervention accuracy (Geiger et al., 2022): interchange intervention accuracy replaces the values of intermediate nodes with those derived with different sets of inputs and evaluates the rate at which these interventions have an identical effects on the output of the abstract model on and the abstracted output of the concrete model.

If we restrict the set of allowable $\alpha$ and $\gamma$ functions, for example to linear mappings alone, Axioms 1 through 4 can distinguish between the two hypotheses. In particular, as no more than two points on the curve $(x, 1/x)$ can be collinear, Axiom 1 cannot hold with $\epsilon = 0$ on any distribution with more than four points in its support when we restrict $\alpha$ and $\gamma$ to be linear.

In the general case, with unrestricted $\alpha$ and $\gamma$, the interpretations are again indistinguishable. However to obtain this result, we must conduct meaningful computation in the abstraction and concretization functions, e.g. $\alpha_4(x) = \gamma_4(x) = \frac{1}{x}$; while not directly evaluated by the axioms, any analyst with a bias towards keeping complexity in the interpretation would prefer

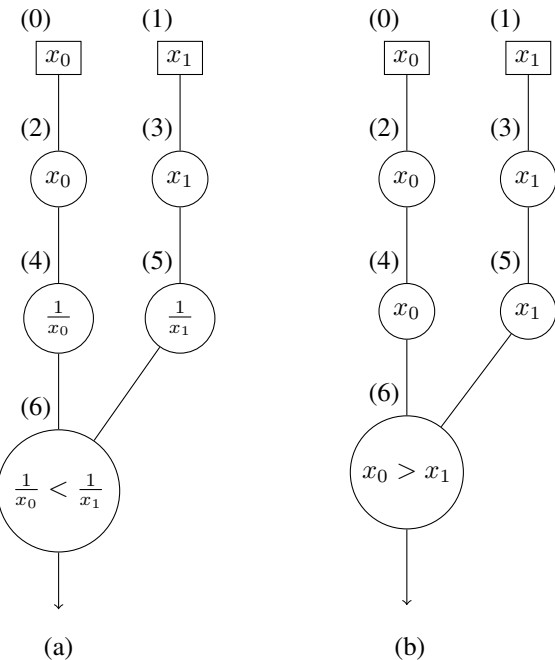

Figure 4: Extensionally equivalent mechanistic interpretations considered by Scheurer et al. (2023). Model (a) is the concrete model and the ground truth interpretation. Model (b) is the abstract model.

the ground-truth interpretation.

## D. More Details on Mechanistic Interpretability Analysis of the 2-SAT Model

The decomposition of the original Transformer 2-SAT solver is shown in Figure 5.

The abstraction operators from our analysis are shown in Listings 1 and 2 and the concretization operators are shown in Listings 3 and 4.

### D.1. First Block as Parser

**Decomposed Attention Analysis.** The parsing behavior in the first block cannot be fully described by token-to-token attention, and hence, we further decompose the standard QK-decomposition of Elhage et al. (2021) to account for positional factors as well. In a Transformer, each token's initial embedding is the sum of a positional embedding and a token embedding; hence, rather than viewing attention as from destination embeddings to source embeddings, we can equivalently express the pre-softmax scores as the sum of four sets of preferences—token-to-token attention, token-to-position attention, position-to-token attention, and position-to-position attention. In particular we can decompose the first block's self-attention mechanism as follows.[5] Given token $t_{\text{src}}$ in position $p_{\text{src}}$ to token $t_{\text{dst}}$ in position $t_{\text{dst}}$, the pre-softmax score is

$$
\begin{aligned}
&\left(t_{\text{src}}{}^T W_E{}^T + p_{\text{src}}{}^T W_{POS}{}^T\right) W_K{}^T W_Q \left(W_E t_{\text{dst}} + W_{POS} p_{\text{dst}}\right) \\
&= \left(t_{\text{src}}{}^T W_E{}^T + p_{\text{src}}{}^T W_{POS}{}^T\right) W_{QK} \left(W_E t_{\text{dst}} + W_{POS} p_{\text{dst}}\right) \\
&= t_{\text{src}}{}^T W_E{}^T W_{QK} W_E t_{\text{dst}} + t_{\text{src}}{}^T W_E{}^T W_{QK} W_{POS} t_{\text{src}} + \\
&\quad p_{\text{src}}{}^T W_{POS}{}^T W_{QK} W_E t_{\text{dst}} + p_{\text{src}}{}^T W_{POS}{}^T W_{QK} W_{POS} p_{\text{dst}} \\
&= t_{\text{src}}{}^T W_{QK}^{\text{tok}\to\text{tok}} t_{\text{dst}} + t_{\text{src}}{}^T W_{QK}^{\text{tok}\to\text{pos}} p_{\text{src}} + p_{\text{src}}{}^T W_{QK}^{\text{pos}\to\text{tok}} t_{\text{dst}} + p_{\text{src}}{}^T W_{QK}^{\text{pos}\to\text{pos}} p_{\text{dst}},
\end{aligned}
$$

where $W_E$ and $W_{POS}$ are the token and positional embedding matrices, $W_{QK}$ is the QK matrix, $t_{\text{src}}$ and $t_{\text{dst}}$ are the source and destination tokens, while $p_{\text{src}}$ and $p_{\text{dst}}$ are source and destination positions.

---

[5]We use the QK circuit notation of Elhage et al. (2021)

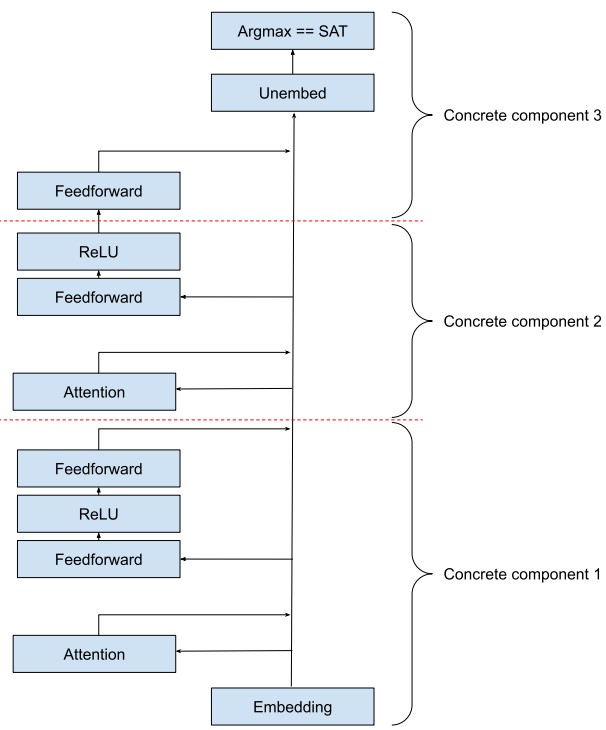

Figure 5: Breakdown of the concrete model into concrete components, corresponding to the components $d_t[i]$.

Our observations in Figures 1 and 6 suggest that the tokens relevant to the final classification decision, are in position $4i + 2$ for $0 \leq i < 10$, which, given the structure of the inputs to the model, are the second tokens of each clause of the form $x_i$ or $\neg x_i$ for $0 \leq i < 5$. If we restrict our view to destination tokens at this position, we can express the effects of the four components of the first block's QK circuit as shown in Figure 7. With the somewhat odd exception of the first clause, and, in particular, a first clause with second literal $x_0$ or $\neg x_0$, attention strongly deprioritizes the parentheses but has weak preferences for the source token otherwise. Outside of destination tokens $x_0$ and $\neg x_0$, preferences for source position are fairly consistent, and hence, the core effect is a combination of position-to-position preferences (see the bottom right of Figure 7) and a preference against punctuation (the parentheses tokens '(' and ')'). This can be seen in the token-to-position preferences in the bottom left, with an exception in the case of token 2, which may behave differently as all tokens to the left belong to the first clause, limiting the importance of learning more specific attention patterns. A general preference against punctuation can be seen in the token-to-position preferences (top left) as well. For the first several clauses, positional preferences encode a preference for the positions where parentheses occur, open parentheses in particular; however, the preference against parentheses shown in the token-to-position attention scores (corresponding to $W_{QK}^{\text{tok} \to \text{pos}}$) counteract that effect to result in a net effect of attention to the first literal of the clause.

**Distributional Attention Analysis.** To illustrate how these four components of the QK circuit work together on typical formulas, we can use the values of the four QK matrices to compute the attention scores in expectation given a uniformly distributed choice of literals. To compute the attention scores in expectation, we only consider destination positions $p$ ($p = 4i + 2$ for some $i$). Call the token in that position, $t$, and the full embedding passed to attention, $e$. To compute the expected pre-softmax attention score on position $p' \leq p$ (defining $t'$ and $e'$ similarly), we first check if $p'$ contains punctuation. If $p' \equiv 0 \pmod 4$, $t'$ is '(' and if $p' \equiv 3 \pmod 4$, $t'$ is ')'. For such $t$ and $t'$ we can use our decomposition

```
1  alpha_0 = identity
2
3  def clause_representation(literal_l, literal_r):
4      # Generate the input tensor for the formula with 10 copies of the clause
5      inputs = to_toks([Or(literal_l, literal_r)] * 10)
6
7      embeds = model.embed(inputs)
8      attn_out = embed + model.blocks[0].attention(
9          embeds,
10         mask=gen_attention_mask([4*i+2: clause_positions_and_parens(i)])
11     )
12     block_1_out = attn_out + model.blocks[0].mlp(attn_out)
13
14     # We derive the canonical representation by averaging across second literal positions
15     return mean(block_1_out[4*clause_idx+2] for clause_idx in range(10))
16
17 clause_representations = {
18     Or(l,r): clause_representation(l, r)
19     for l in literals
20     for r in literals
21 }
22
23 def alpha_1(block_1_output):
24     second_literal_outputs = [block_1_outputs[4*i+2] for i in range(10)]
25     # Calculate cosine similarities between the actual representations output at
26     # the second variable positions and the canonical clause
27     # representations, select the clauses which best match
28     return
       [argmax_cosine_sims(out, clause_representations) for out in second_literal_outputs]
```

Listing 1: Abstraction operators for the first block in Python pseudocode.

```
1  def alpha_2(residual_and_mlp_output, threshold=0.5):
2      mlp_output = residual_and_mlp_output[1]
3      return [mlp_output[i] >= threshold for i in evaluating_neurons]
4
5  alpha_3 = identity
```

Listing 2: Abstraction operators for the second block in Python pseudocode.

into the four sets of preferences to calculate an expected score:

$$
\begin{aligned}
\mathbb{E}_{t,t'} e'^T W_{QK} e &= \mathbb{E}_{t',t} \left[ t'^T W_{QK}^{\text{tok}\to\text{tok}} t + t'^T W_{QK}^{\text{tok}\to\text{pos}} p + p'^T W_{QK}^{\text{pos}\to\text{tok}} t + p'^T W_{QK}^{\text{pos}\to\text{pos}} p \right] \\
&= \mathbb{E}_t t'^T W_{QK}^{\text{tok}\to\text{tok}} t + t'^T W_{QK}^{\text{tok}\to\text{pos}} p + \mathbb{E}_t p'^T W_{QK}^{\text{pos}\to\text{tok}} t + p'^T W_{QK}^{\text{pos}\to\text{pos}} p \\
&= \frac{1}{|lit|} \sum_{t \in lit} W_{QK}^{\text{tok}\to\text{tok}}{}_{t',t} + t'^T W_{QK}^{\text{tok}\to\text{pos}} p + \\
&\quad \frac{1}{|lit|} \sum_{t \in lit} W_{QK}^{\text{pos}\to\text{tok}}{}_{p',t} + p'^T W_{QK}^{\text{pos}\to\text{pos}} p
\end{aligned}
$$

where we overload notation and use $t$, $t'$ as tokens, indices, and the corresponding one-hot representations, and similarly for the positions $p$ and $p'$ and where $lit$ is the set of literals.

If $p'$ does not contain punctuation, then we know that $t'$ is a variable ($x_i$ or $\neg x_i$ for some $i$), and similarly for $t$. Using

```
1  gamma_0 = identity
2
3  # Mean output from the first transformer block, by position, on the training set
4  mean_block_1_out = mean(model.intermediate_outputs("block_1", x) for x in train)
5
6  # Constant on everything but second token positions
7  def gamma_1(clauses):
8      output = mean_block_1_out.copy()
9      # Substitute the corresponding canonical clause representation
10     for i, clause in enumerate(clauses):
11             output[4*i+2] = clause_representations[clause]
12     return output
```

Listing 3: Concretization operators for the first block in Python pseudocode.

```
1  # Mean residual from attention for the readout token on the training set
2  mean_attn_out
       = mean(model.intermediate_outputs("attention_block_2", x)[-1] for x in train)
3
4  # Output a constant residual term, and map predicted activations to
5  # a large constant value
6  def gamma_2(activation_model_outputs, high_activation=2):
7      mlp_out_concretization = torch.zeros(mlp_width)
8      for i, model_out in zip(evaluating_neurons, activation_model_outputs):
9          if model_out:
10             mlp_out_concretization[i] = high_activation
11
12     return mean_attn_out, mlp_out_concretization
13
14 gamma_3 = identity
```

Listing 4: Concretization operators for the second block in Python pseudocode.

that, we can use our decomposition into the four sets of preferences to calculate an expected score:

$$
\begin{aligned}
\mathop{\mathbb{E}}_{t,t'} e'^T W_{QK} e &= \mathop{\mathbb{E}}_{t',t} \left[ t'^T W_{QK}^{\text{tok}\to\text{tok}} t + t'^T W_{QK}^{\text{tok}\to\text{pos}} p + p'^T W_{QK}^{\text{pos}\to\text{tok}} t + p'^T W_{QK}^{\text{pos}\to\text{pos}} p \right] \\
&= \mathop{\mathbb{E}}_{t',t} t'^T W_{QK}^{\text{tok}\to\text{tok}} t + \mathop{\mathbb{E}}_{t'} t'^T W_{QK}^{\text{tok}\to\text{pos}} p + \mathop{\mathbb{E}}_{t} p'^T W_{QK}^{\text{pos}\to\text{tok}} t + p'^T W_{QK}^{\text{pos}\to\text{pos}} p \\
&= \frac{1}{|lit|^2} \sum_{t,t'\in lit} W_{QK}^{\text{tok}\to\text{tok}}{}_{t',t} + \frac{1}{|lit|} \sum_{t'\in lit} W_{QK}^{\text{tok}\to\text{pos}}{}_{t',p} + \\
&\quad \frac{1}{|lit|} \sum_{t\in lit} W_{QK}^{\text{pos}\to\text{tok}}{}_{p',t} + p'^T W_{QK}^{\text{pos}\to\text{pos}} p
\end{aligned}
$$

After taking the softmax of the resulting values, the result is shown in Figure 8.

**Worst-case Attention Analysis.** We can also show that the parsing behavior occurs by a worst-case analysis of the attention scores, i.e. what is the minimum weight from attention to the first token of the clause and to the clause as a whole? This allows us to dispense with any distributional assumptions, and to validate that the inconsistent preferences for particular literals (i.e. the differences in preferences when the destination token is $x_0$). Given the definition of softmax,

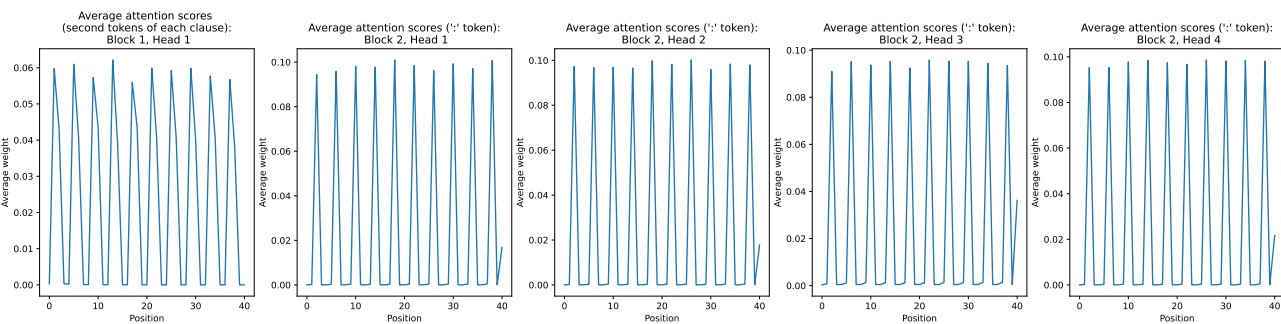

Figure 6: Average attention scores, grouped by source token position, for all heads, calculated over the test set. In the first block, we further average across destination token positions restricted to positions $4i + 2$, where $0 \leq i < 10$ is the clause index. For the second block, we only consider the readout token as the destination.

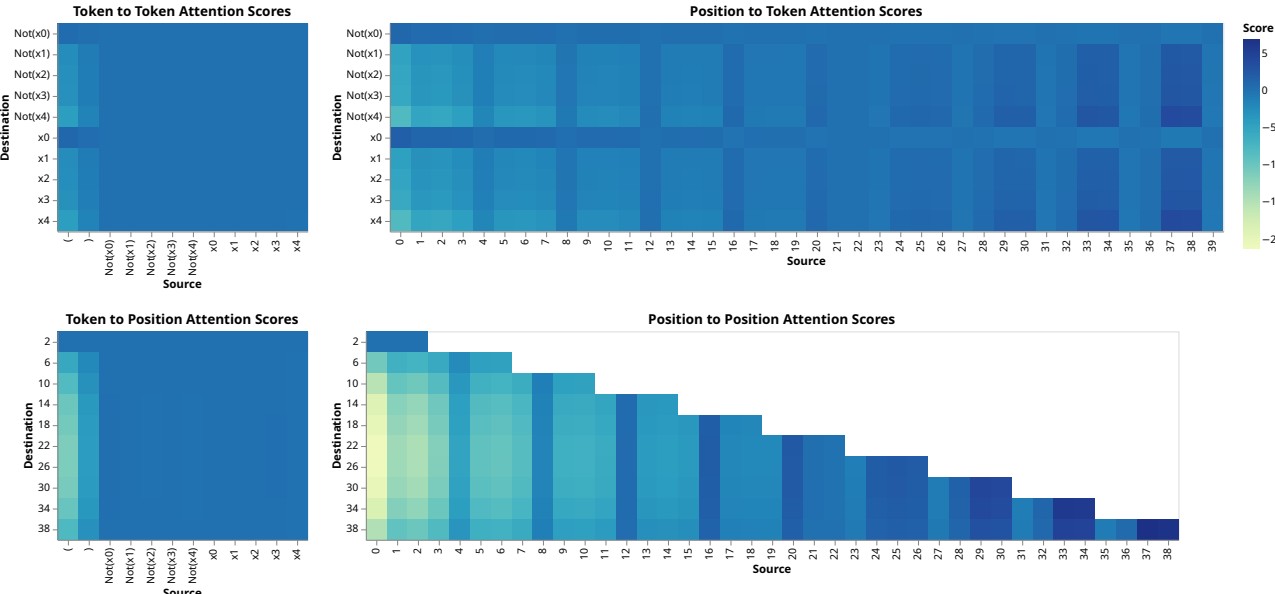

Figure 7: Decomposed QK circuit of the first block's attention mechanism; the subset where the destination token is the second variable of a clause is shown.

we can compute the minimum weight on the first token of clause $i$ by the second token of clause $i$ as follows:

$$\min_{\phi} \mathrm{score}^{\mathrm{softmax}}_{4i+1, 4i+2}(\phi)$$

$$= \min_{t \in tok_{4i+2}} \min_{\{\phi | \phi_{4i+2}=t\}} \frac{e^{\mathrm{score}_{4i+1, 4i+2}(\phi)}}{\sum_{j \leq 4i+2} e^{\mathrm{score}_{j, 4i+2}(\phi)}}$$

$$\leq \min_{t \in tok_{4i+2}} \frac{e^{\min_{\{\phi | \phi_{4i+2}=t\}} \mathrm{score}_{4i+1, 4i+2}(\phi)}}{e^{\min_{\{\phi | \phi_{4i+2}=t\}} \mathrm{score}_{4i+1, 4i+2}(\phi)} + \sum_{j \leq 4i \lor j = 4i+2} e^{\max_{\{\phi | \phi_{4i+2}=t\}} \mathrm{score}_{j, 4i+2}(\phi)}}$$

where $tok_i$ is the set of all possible tokens in position $i$ and where $\mathrm{score}_{s,d}(\phi)$ and $\mathrm{score}^{\mathrm{softmax}}_{s,d}(\phi)$ refer to the pre- and post-softmax attention scores with destination token in position $d$ and source token in position $s$ and where the input formula is $\phi$. For the full clause, the approach is similar, except we add the term $e^{\min_{\{\phi | \phi_{4i+2}=t\}} \mathrm{score}_{4i+2, 4i+2}(\phi)}$ to the numerator.

Now, we can derive the minimal and maximal scores for and $s$ and $d$ using our decomposition of the QK circuit:

$$\min_{\{\phi | \phi_d = t\}} \mathrm{score}_{s,d}(\phi) = W_{QK}^{\mathrm{pos} \rightarrow \mathrm{pos}}{}_{s,d} + \min_{t' \in tok_s} W_{QK}^{\mathrm{tok} \rightarrow \mathrm{pos}}{}_{t',d} + W_{QK}^{\mathrm{pos} \rightarrow \mathrm{tok}}{}_{s,t} + \min_{t' \in tok_s} W_{QK}^{\mathrm{tok} \rightarrow \mathrm{tok}}{}_{t',t}$$

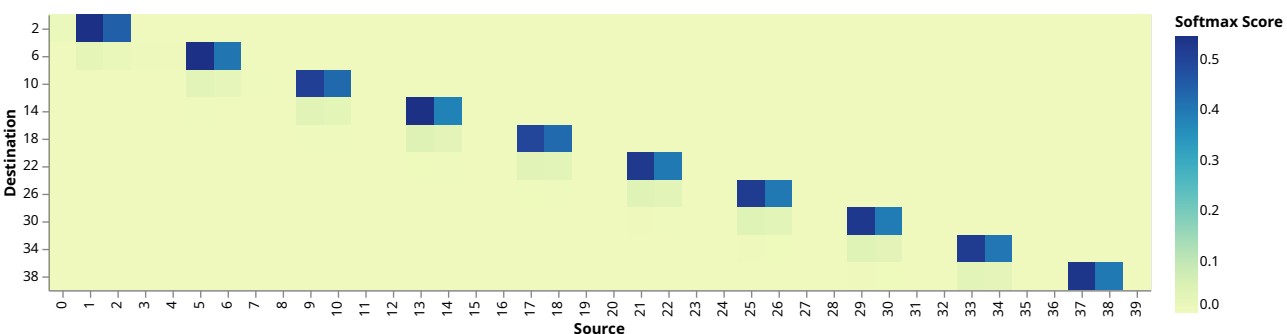

Figure 8: Softmax of expected attention scores by position for the second token of each clause, showing that for each second token position, attention on literal positions in the corresponding clause are expected to dominate, consistent with our interpretation of the first block's attention mechanism as a parser.

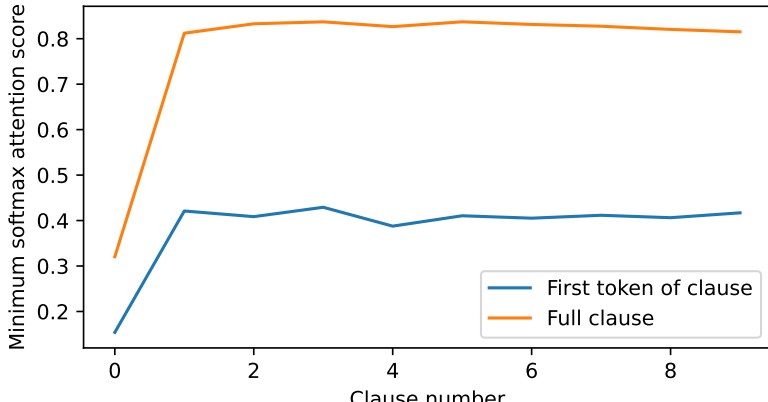

Figure 9: Minimum post-softmax attention weights for each clause.

and similarly for the maximum (noting that if $s = d$, we must have $t' = t$ as well).

The results are shown in Figure 9, and demonstrate that regardless of the choice of formula, the majority of attention will be placed on the clause as a whole, except in the case of the first clause, in which case any additional attention is to the first '(' token, which contains no useful information.

Now, we can perform similar analysis on the attention of the ':' token. We observe in Figure 1 that the attention of the ':' token in the first block is near uniform. To see why this occurs, see Figure 10, which fully describes the first-block attention of ':'. These are derived from the four QK matrices using the known position $c = 40$ of the token. The attention scores are consistently very small and have very little variation. As above, we can use this set of preferences to calculate a lower bound on the attention paid by the ':' token to any individual token:

$$\min_{i,\phi} \text{score}^{\text{softmax}}_{i,40}(\phi) \leq \frac{e^{\min_{i,t} \text{score}^{(:)}_{i,t}}}{e^{\min_{i,t} \text{score}^{(:)}_{i,t}} + 40 e^{\max_{i,t} \text{score}^{(:)}_{i,t}}}$$

where $\text{score}^{(:)}_{i,t}$ refers to the ":" token's pre-softmax attention to token $t$ in position $i$. We derive that the minimum attention paid to any token by ':' is approximately 0.0209, and we can similarly show that the maximum attention to any token is $\approx 0.0285$; hence, the ":" token's attention can never vary far from the $\approx 0.0244$ paid by uniform attention.

**Interpretation.** Listing 5 shows our extracted mechanistic interpretation of the first component as Python code.

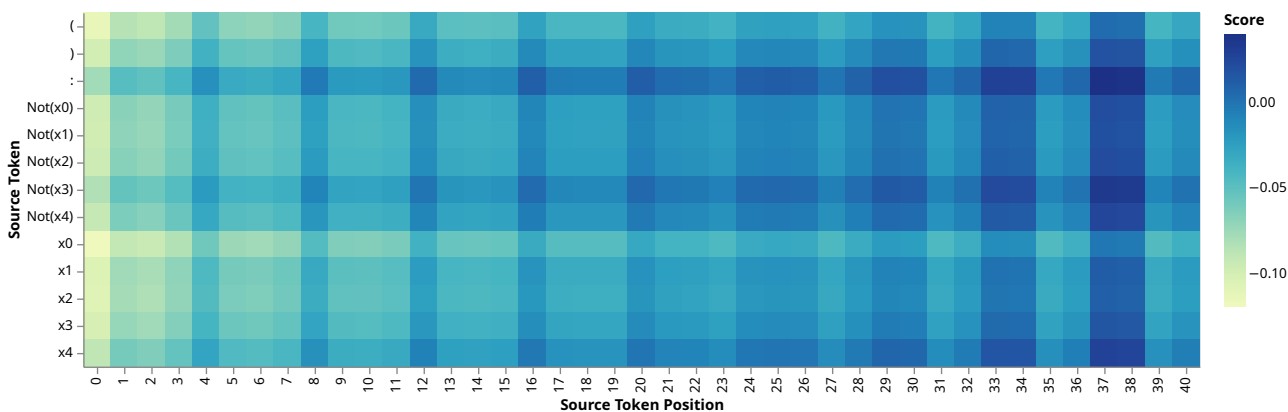

Figure 10: Pre-softmax attention scores by the ':' token to each token, by position and token type.

```python
def parse_clauses(formula: list[tok]) -> list[tuple[lit, lit]]:
    return [(formula[4*i+1], formula[4*i+2]) for i in range(10)]
```

Listing 5: First component's mechanistic interpretation in Python syntax.

### D.2. Second Block as Evaluator

**Abstract Attention Analysis.** We showed in Section 4.2.1 that the behavior of the first block is accurately captured by an interpretation which, after the application of $\gamma_1$, outputs canonical representations for each clause at the second literal positions and is constant at all other positions. Prefix replaceability, in particular, enforces this property directly. Furthermore, we must only study attention with the readout token as the destination to explain the output behavior; hence, we can characterize the behavior of attention in the second block solely by the preferences of each head in the readout token position to the canonical clause representations in the second literal positions, dramatically reducing dimensionality.

In this way, the key behavior of attention in the second block is fully described by Figure 11. Each of the four charts shows the preferences on each clause by the corresponding head; the columns correspond to the first literal of the clause and the rows to the second.

We see that the behavior of attention is nearly identical for a clause and its equivalent mirror image (i.e. the attention score on $(l \vee r)$ is approximately the same as that on $(r \vee l)$); this is as expected given that, logically, $l \vee r = r \vee l$. We can also see that each head prefers clauses containing certain literals (for instance, head 0 prefers negated literals as well as $x_0$ and $x_3$); each clause receives a high score from some head (so no clauses are overlooked by the attention mechanism).

While we might read far more into these patterns, this is a mistake in this case. In this instance, the behavior of attention is, in fact, not relevant to the underlying algorithm and serves only to obscure the underlying behavior. As discussed in Section 4.2.2, we can much better understand attention in concert with the first layer of the MLP; collectively, these components encode the observed evaluating behavior, as discussed in more detail below.

**Identifying the Key Pathway.** We observe that the unembedding vector for UNSAT is nearly exactly the negative of the unembedding vector for SAT ($\|W_U^{\text{SAT}} + W_U^{\text{UNSAT}}\| \approx 6.2 \times 10^{-6}$); hence, the SAT logit and the UNSAT logit are almost exactly negatives of each other; furthermore, the model's output token is always SAT or UNSAT. Across all formulas in the analysis dataset, the minimum value of the logit associated with the prediction (either SAT or UNSAT) is slightly over 0, while while the maximum logit on any other token across the dataset is below -9. Hence, we can consider the behavior of the SAT logit exclusively.

Next, we show that the key pathway that determines classification passes through the second block's MLP. For an input formula, the effect of the second block's attention mechanism through the residual connection on the SAT logit simply the dot product of the post-attention embedding in the readout token position with $W_U^{\text{SAT}}$; as we can see in Figure 12, this value is fairly consistent across the dataset and always negative; so a positive SAT logit and hence a SAT classification *must* depend on the action of the MLP.

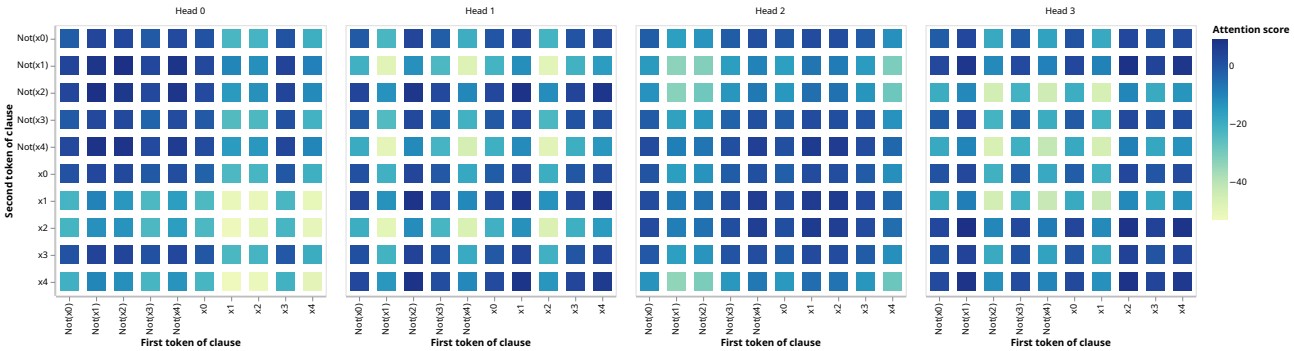

Figure 11: Second-block abstract attention preferences.

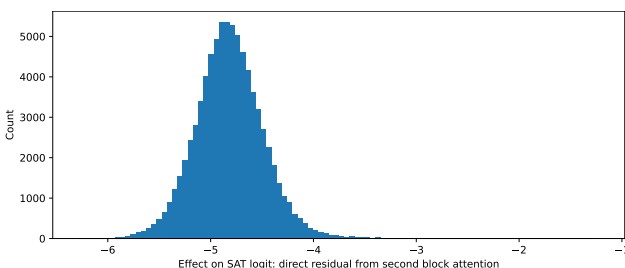

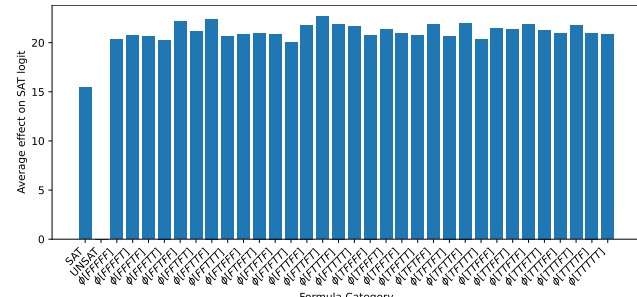

Figure 12: Effect of the residual connection from the second block's attention mechanism on the SAT logit. x-axis shows the effect on the SAT logit and y-axis is the number of instances in the analysis test dataset.

Figure 13: Average total effect on the SAT logit across all relevant neurons from SAT, UNSAT, and formulas satisfiable with particular assignments to the variables.

**Evaluation in the Hidden Neurons.** As discussed in Section 4.2.2, only 34 hidden neurons have a significant effect on classification; hence, we focus on the behavior of these neurons. As observed in Figure 12, the effect of the residual stream from the second block attention on the SAT logit reduces the SAT logit by less than 6 units for all formulas in the analysis test dataset. Furthermore, as observed in Section 4.2.2, the model always predicts SAT or UNSAT, and the SAT and UNSAT logits are tied (in particular, they are negatives). Hence, the model will predict SAT when the collective effect of these 34 neurons is to increase SAT by > 6 units. We have observed that the expected behavior of these neurons is strongly dependent on *which* assignments to the variables satisfy the formula $\phi$.

A natural question to ask is whether, whether these neurons will collectively increase the SAT logit sufficiently to output SAT on SAT formulas regardless of which assignment to the variables satisfies the formulas. Calculating this in the same way as we do for an individual neuron in Figure 3, we obtain Figure 13, which shows that, indeed, the collective effect of the evaluating neurons (recall that we refer to the 34 relevant neurons as evaluating neurons) correctly identifies SAT formulas regardless of *how* the formulas are satisfiable. It may seem odd that the average effect of SAT is significantly below the minimum of the effects for formulas satisfiable with any given assignment, as any SAT formula belongs to one such category; this is an artifact of the fact that no neuron has a large effect for all SAT formulas, and that each specializes in a subset.

Now, each of these neurons has a large positive (> 2.9) output coefficient (recall that the output coefficient of a neuron is the weight in the weighted sum expression constructed by composing the output layer of the MLP and the unembedding matrix projected to the SAT logit), so an activation above 2 units on any of the evaluating neurons is enough to force SAT, ignoring all other neurons; hence, the model outputs SAT if *any* evaluating neuron activates sufficiently strongly. In this sense, the effect of the final output components (the MLP's output layer, the unembedding matrix, and the residual from attention) can be viewed as an OR operation.

However, in reality, individual neurons do not consistently reach activations of $\geq 2$ (for example, see Figure 3, in which the average output value for formulas satisfiable with assignment TFFFF is approximately 1.5), and multiple neurons

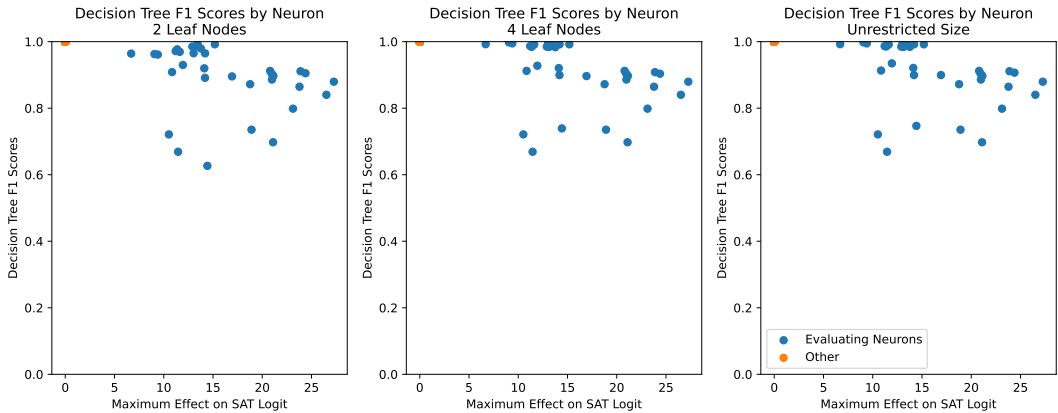

Figure 14: F1 scores of decision tree interpretations classifying high (> 0.5) activation of MLP hidden neurons.

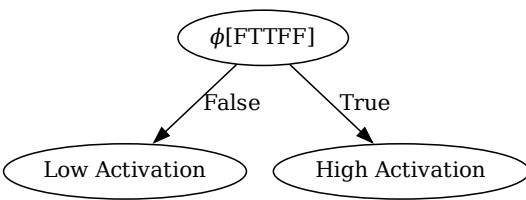

Figure 15: Decision tree interpretation for unit 29 of the second block MLP's hidden layer, maximum leaf nodes: two.

recognize a given SAT formula. Hence, moderate activation may be necessary across several neurons to predict SAT in some cases; this may explain the redundancy in evaluation behavior between neurons seen in Appendix G. For simplicity in our interpretation (in particular, to allow replacing a numerical weighted sum with Boolean operations) we handle this by using varying thresholds between the corresponding $\alpha$ and $\gamma$ functions for the MLP's hidden layer (in particular, $\gamma_2$ outputs an activation of 2 for neurons whose interpretation predicts high activation, while 0.5 is considered a high activation in the case of $\alpha_2$); as noted in the paragraph titled **Hidden Layer of Second Block** in Section 4.2.3, if $\gamma_2$ and $\alpha_2$ both use the lower threshold, replacing the neural network components with our interpretation no longer has a minimal effect on classification.

**Interpreting Neurons via Decision Trees.** We observe that each assignment to the variables has *some* neuron for which formulas satisfiable with that assignment result in a significant increase to the SAT logit on average, as we'd expect if the model was implementing the natural exhaustive enumeration algorithm for satisfiability checking. In particular, every assignment $a$ has some neuron which increases the SAT logit by at least 2.9 units on average on formulas where $a$ is a satisfying assignment; however, the average-case analysis hides the complexity of the behavior of these neurons. We can see this by noting that for every relevant neuron that on average demonstrates high activation for formulas satisfiable with some assignment $a$, we are able to find some formula satisfiable with $a$ which fails to result in a sufficient activation for the neuron.

We'll focus on the general decision tree classifiers discussed in Section 4.2.2 here. We train decision tree classifiers on the thresholded activations of each evaluating neuron on the analysis training set; this allows us to learn an arbitrary Boolean function in the features $\phi[...]$. We observe that these decision trees do indeed reliably predict activation. In particular, even very limited-size decision trees are reasonably strong predictors of the behavior of the neurons; we limit our decision trees to four leaf nodes in our experiments. Figures 15, 16, and 17 illustrate the effect on the decision trees as we lift this constraint. Figure 14 shows that more complex decision trees are not significantly better predictors of the behavior of the neurons.

**Implementation of Decision Trees by the Model.** To see how the model implements the decision trees, we'll consider the effect of the composition of the second block MLP's hidden layer with the attention mechanism on the canonical representa-

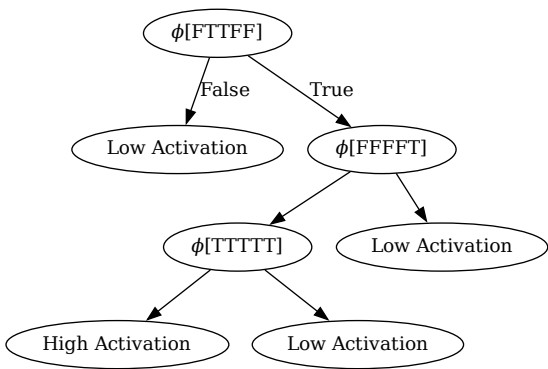

Figure 16: Decision tree interpretation for unit 29 of the second block MLP's hidden layer, maximum leaf nodes: four.

tions of the clauses, using the abstract attention approach discussed earlier. As the pre-softmax attention score by each head on the second literal of a clause is a fixed value for each clause (recall that $\gamma_1$ leaves the representation of the readout token constant), we can express the effect of attention precisely in terms of the number of occurrences of each clause, as we show next.

For head $h$ and clause $l \vee r$, let the pre-softmax attention score paid by $h$ to the clause be $w^h_{l\vee r}$. Moreover, the output of the OV circuit is likewise a fixed vector for each clause, call it $v^h_{l\vee r}$. Say that our formula contains a list of $n$ clauses ($n = 10$ for all our experiments) where the $i^{\text{th}}$ clause is $l_i \vee r_i$. Then, the attention paid to the $i^{\text{th}}$ clause by head $h$ is:

$$\text{softmax}([w^h_{l_j \vee r_j}]_{j\in[1..n]})_i = \frac{e^{w^h_{l_i \vee r_i}}}{\sum_{j=1}^{n} e^{w^h_{l_j \vee r_j}}}$$

and hence the output of the attention mechanism (call it $o$) at the readout token is, by rearranging terms,

$$e_: + \sum_{h=1}^{m} \sum_{i=1}^{n} \text{softmax}([w^h_{l_j \vee r_j}]_{j\in[1..n]})_i v^h_{l_i \vee r_i} = e_: + \sum_{h=1}^{m} \frac{\sum_{l\in lit, r\in lit} e^{w^h_{l\vee r}} count_{l\vee r}(\phi) v^h_{l\vee r}}{\sum_{l\in lit, r\in lit} e^{w^h_{l\vee r}} count_{l\vee r}(\phi)}$$

where $count_{l\vee r}(\phi)$ is the number of occurrences of $l \vee r$ in the formula $\phi$ and where $m$ is the number of attention heads and where $e_:$ is the fixed representation of the readout token ':'.

Now, consider the action of the MLP's hidden layer on this value: in particular, consider the effect on a specific neuron $n$, where $w_n$ is the corresponding column of the MLP's input weight matrix and $b_n$ is the corresponding bias value.

The pre-activation score for $n$ will then be:

$$b_n + w_n^T o = b_n + w_n^T \left( e_: + \sum_{h=1}^{m} \frac{\sum_{l\in lit, r\in lit} e^{w^h_{l\vee r}} count_{l\vee r}(\phi) v^h_{l\vee r}}{\sum_{l\in lit, r\in lit} e^{w^h_{l\vee r}} count_{l\vee r}(\phi)} \right)$$

$$= b_n + w_n^T e_: + \sum_{h=1}^{m} \frac{\sum_{l\in lit, r\in lit} e^{w^h_{l\vee r}} count_{l\vee r}(\phi) w_n^T v^h_{l\vee r}}{\sum_{l\in lit, r\in lit} e^{w^h_{l\vee r}} count_{l\vee r}(\phi)}$$

$$= c_n + \sum_{h=1}^{m} \frac{\sum_{l\in lit, r\in lit} c^{h,n}_{l\vee r} count_{l\vee r}(\phi)}{\sum_{l\in lit, r\in lit} d^h_{l\vee r} count_{l\vee r}(\phi)}$$

where we define the following constants: $c_n = b_n + w_n^T e_:$, $d^h_{l\vee r} = e^{w^h_{l\vee r}}$, $c^{h,n}_{l\vee r} = d^h_{l\vee r} w_n^T v^h_{l\vee r}$.

Note that the numerator and denominator of the fraction describing the contribution of head $h$ are both linear functions of the number of occurrences of each clause in the formula.

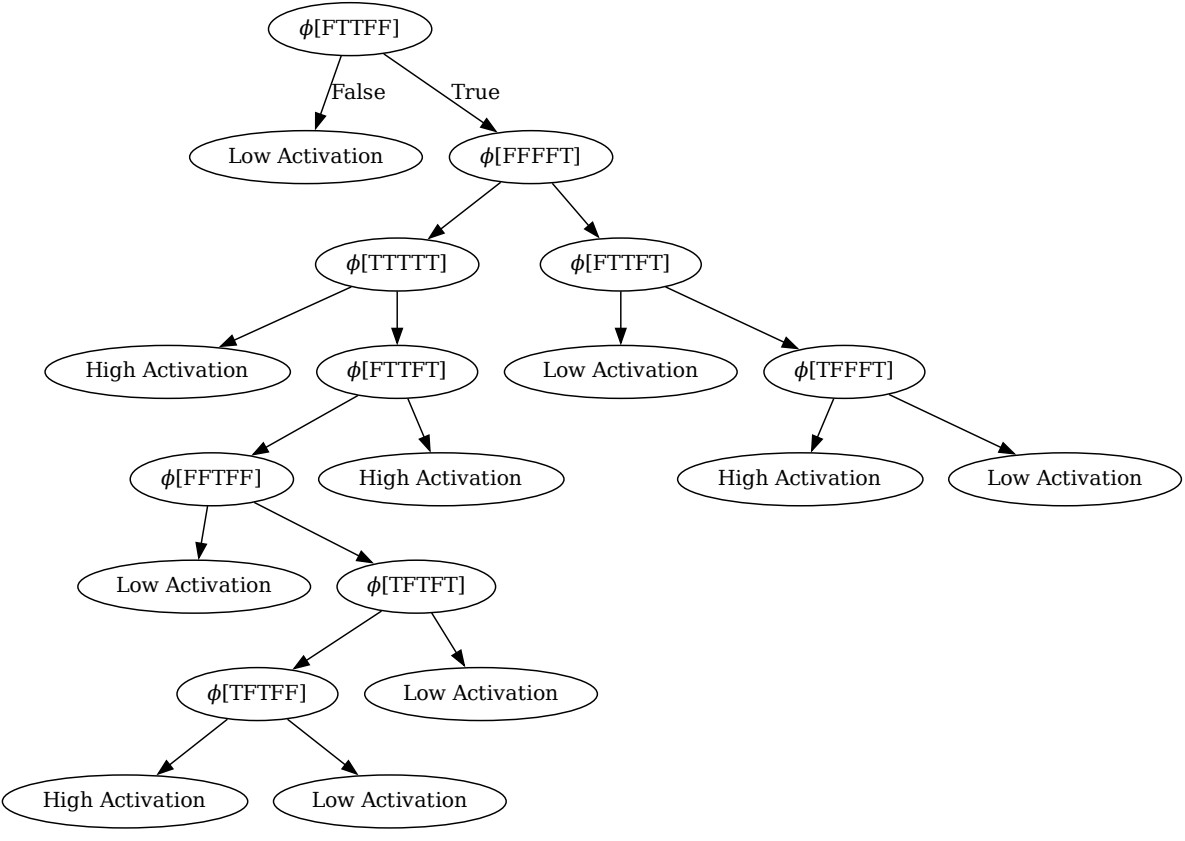

Figure 17: Decision tree interpretation for unit 29 of the second block MLP's hidden layer, unlimited leaf node count.

We'll briefly describe how $\phi[a]$ and $\neg\phi[a]$ can be implemented with a single neuron for an arbitrary assignment $a$ to the variables in this formulation (in particular, we can ensure activation if and only if the condition that $\phi$ is satisfiable by $a$ or not satisfiable by $a$ holds), and then analyze the behavior of a neuron with an interpretation of this form.

As the formula $\phi$ is in conjunctive normal form, $\phi[a]$ is true if and only if none of the clauses $l \vee r$ in $\phi$ evaluate to false given the assignment $a$ to the variables (in particular, this holds if neither $l$ nor $r$ holds given $a$). We can then implement $\phi[a]$ in neuron $n$ if we let $d^h_{l \vee r}$ be constant, $c^{h,n}_{l \vee r} = 0$ for $l \vee r$ which are true given assignment $a$, and let $c^{h,n}_{l \vee r} = -\infty$ for $l \vee r$ which are false given $a$, and then let $c_n$ be a large positive number. Then, if $\phi[a]$, $count_{l \vee r}(\phi) = 0$ for $l \vee r$ unsatisfiable given $a$, so the output is $c_n$. Otherwise, if $\neg\phi[a]$, $count_{l \vee r}(\phi) > 0$ and so the output is $-\infty$. After applying ReLU, the activation of unit $n$ is a large positive number when $\phi[a]$ and zero otherwise. Similarly, we can implement $\neg\phi[a]$ by assigning $c_n$ to a negative number and letting $c^{h,n}_{l \vee r} = \infty$ for $l \vee r$ which are false given $a$.

Now, suppose that we want to implement some DNF expression $e$ in the $\phi[...]$ without negation of any $\phi[...]$. Say that $e$ contains $k$ conjuncts $e_i$ for $1 \leq i \leq k$ and that the attention mechanism has at least $k$ heads. We can extend the ideas in the implementation of $\phi[a]$ to one for $e$ as follows: let $c_n = k$; now, consider head $h$ with corresponding conjunct $e_i$ as in the $\phi[a]$ case, let $c^{h,n}_{l \vee r} = 0$ for $l \vee r$ which are not true for every assignment $a$ appearing in $e_i$. As in that case, $\phi$ satisfies $e_i$ if and only if no clause violates the constraint (following from the assumption that $e_i$ is a conjunction of $\phi[a_j]$ which appear without negation, as, for an $e_i$ of this form to be satisfied, each clause in $\phi$ must be true for each assignment $a_j$). Hence, if we let $c^{h,n}_{l \vee r} = -1$ for $l \vee r$ which violate $e_i$, let $d^h_{l \vee r} = 1$ for such $l \vee r$ and $d^h_{l \vee r}$ some negligible value for $l \vee r$ which do not violate the condition, the contribution of head $h$ to the expression is 0 if $e_i$ is satisfied and $-1$ otherwise (as the numerator will be $-v$ and the denominator will be $v + \epsilon$ where $v$ is the number of clauses violating the condition). Hence, if all conjuncts $e_i$ fail to be satisfied, the out-

```python
def evaluate_satisfiability(
    clauses: list[tuple[lit, lit]],
    neuron_interpretations: list[Callable[[list[tuple[lit, lit]]], bool]]
) -> list[bool]:
    return [
        neuron_interp(clauses)
        for neuron_interp in neuron_interpretations]

def predict_satisfiability(satisfiabilities: list[bool]) -> bool:
    return any(satisfiabilities)
```

Listing 6: Second block's mechanistic interpretation. First, it applies a series of functions, namely, the neuron interpretations, to the parsed clauses from the first block (`evaluate_satisfiability`) and then, it applies an OR operation (`predict_satisfiability`).

```python
def abstract_model(formula: list[tok]) -> bool:
    return predict_satisfiability(
            evaluate_satisfiability(
                parse_clauses(formula)))
```

Listing 7: Mechanistic interpretation of full model; see Listings 5 and 6 for the component interpretations.

put will be 0, else, some $e_i$ must be satisfied, so the output will be at least $k-(k-1)=1$, and, hence, the neuron will activate.

**Interpretation.** Listing 6 shows our derived mechanistic interpretation for the second block. Listing 7 shows the mechanistic interpretation for the entire model.

## E. Importance of Prefix Axioms: Theoretical Evidence

While it may appear that each componentwise axiom implies the corresponding prefix axiom, for instance, that Axiom 2 implies Axiom 1, this is not the case. While it is possible to derive a bound, the resulting bound is extremely weak, and, for Axioms 3 and 4, we cannot derive even such a weak bound without additional assumptions.

### E.1. Equivalence Axioms

Suppose that Axiom 2, component equivalence, holds with a fixed $\epsilon_0$. We will show that prefix equivalence (Axiom 1) holds at component $i$ with an $\epsilon$ of $i\epsilon_0$; if the number of components $l$ is at least $\lceil \frac{1}{\epsilon_0} \rceil$ the bound becomes useless.

Let $PE_i$ be the event that $x \sim D$ satisfies the prefix equivalence condition for component $i$, i.e. that $\alpha_i \circ d_t[:\ i+1](x) = d_h[:\ i+1] \circ \alpha_0(x)$ and and let $CE_i$ be the event that $x$ satisfies the component equivalence condition for component $i$, i.e. that $\alpha_i \circ d_t[:\ i+1](x) = d_h[i] \circ \alpha_{i-1} \circ d_t[:\ i](x)$. Call $Pr[PE_i]$ as $p_i$.

Now,

$$p_{i+1} = \Pr_{x\sim\mathcal{D}}[\alpha_{i+1} \circ d_t[:\ i+2](x) = d_h[:\ i+2] \circ \alpha_0(x)]$$

$$\geq \Pr_{x\sim\mathcal{D}}[\alpha_{i+1} \circ d_t[:\ i+2](x) = d_h[i+1] \circ \alpha_i \circ d_t[:\ i+1](x) \land d_h[i+1] \circ \alpha_i \circ d_t[:\ i+1](x) = d_h[:\ i+2] \circ \alpha_0(x)]$$

$$\geq \Pr_{x\sim\mathcal{D}}[\alpha_{i+1} \circ d_t[:\ i+2](x) = d_h[i+1] \circ \alpha_i \circ d_t[:\ i+1](x) \land \alpha_i \circ d_t[:\ i+1](x) = d_h[:\ i+1] \circ \alpha_0(x)]$$

$$= Pr[CE_{i+1} \land PE_i],$$

noting that $\alpha_i \circ d_t[:\ i+1](x) = d_h[:\ i+1] \circ \alpha_0(x)$ implies $d_h[i+1] \circ \alpha_i \circ d_t[:\ i+1](x) = d_h[:\ i+2] \circ \alpha_0(x)$ by the definition of the $d_h[:]$ notation.

Now, we have that

$$
\begin{aligned}
Pr[CE_{i+1} \wedge PE_i] &= 1 - Pr[\neg(CE_{i+1} \wedge PE_i)] \\
&= 1 - Pr[\neg CE_{i+1} \vee \neg PE_i] \\
&\geq 1 - (Pr[\neg CE_{i+1}] + Pr[\neg PE_i]).
\end{aligned}
$$

Hence, as $Pr[CE_{i+1}] \geq 1 - \epsilon_0$ by component equivalence, we have $p_{i+1} \geq 1 - (\epsilon_0 + (1 - p_i))$ and so $p_{i+1} \geq p_i - \epsilon_0$. We have $p_1 \geq 1 - \epsilon_0$ as prefix equivalence and component equivalence are identical for the first component, and so $p_i \geq 1 - i\epsilon_0$, and we are done.

If we assume that $CE_{i+1}$ and $PE_i$ are independent, we could derive a stronger bound:

$$
Pr[CE_{i+1} \wedge PE_i] = Pr[CE_{i+1}]Pr[PE_i] \geq (1 - \epsilon_0)p_i
$$

from which we can derive $p_i \geq (1 - \epsilon_0)^i$. However, even this stronger bound grows very weak as depth increases, illustrating the need to evaluate a mechanistic interpretation compositionally.

### E.2. Replaceability Axioms

We can only derive a bound in the case that $\gamma_i = \alpha_i^{-1}$ and that all concrete components are invertible. Assume component replaceability, Axiom 4 is satisfied.

Similarly to Appendix E.1 let the $PR_i$ and $CR_i$ be the events that $x \sim D$ satisfies the prefix replaceability and component replaceability conditions for component $i$, i.e. that $t(x) = d_t[i + 1 :] \circ \gamma_i \circ d_h[: i + 1] \circ \alpha_0(x)$ and that $t(x) = d_t[i + 1 :] \circ \gamma_i \circ d_h[i] \circ \alpha_{i-1} \circ d_t[: i](x)$, respectively. By the assumption that all concrete components are invertible, $PR_i$ iff $d_t[: i + 1] = \gamma_i \circ d_h[: i + 1] \circ \alpha_0(x)$ and $CR_i$ iff $d_t[: i + 1] = \gamma_i \circ d_h[i] \circ \alpha_{i-1} \circ d_t[: i](x)$. Call $Pr[PR_i]$ $p_i$.

Now,

$$
\begin{aligned}
p_{i+1} &= \Pr_{x \sim D}[t(x) = d_t[i + 2 :] \circ \gamma_i \circ d_h[: i + 2] \circ \alpha_0(x)] \\
&= \Pr_{x \sim D}[d_t[: i + 2] = \gamma_{i+1} \circ d_h[: i + 2] \circ \alpha_0(x)] \\
&\geq \Pr_{x \sim D}\left[ \begin{array}{l} d_t[: i + 2] = \gamma_{i+1} \circ d_h[i + 1] \circ \alpha_i \circ d_t[: i + 1](x) \wedge \\ \gamma_{i+1} \circ d_h[: i + 2] \circ \alpha_0(x) = \gamma_{i+1} \circ d_h[i + 1] \circ \alpha_i \circ d_t[: i + 1](x) \end{array} \right] \\
&\geq \Pr_{x \sim D}\left[ \begin{array}{l} d_t[: i + 2] = \gamma_{i+1} \circ d_h[i + 1] \circ \alpha_i \circ d_t[: i + 1](x) \wedge \\ d_h[: i + 1] \circ \alpha_0(x) = \alpha_i \circ d_t[: i + 1](x) \end{array} \right] \\
&= \Pr_{x \sim D}\left[ \begin{array}{l} d_t[i + 2 :] \circ d_t[: i + 2] = d_t[i + 2 :] \circ \gamma_{i+1} \circ d_h[i + 1] \circ \alpha_i \circ d_t[: i + 1](x) \wedge \\ d_t[i + 1 :] \circ \gamma_i \circ d_h[: i + 1] \circ \alpha_0(x) = d_t[i + 1 :] \circ \gamma_i \circ \alpha_i \circ d_t[: i + 1](x) \end{array} \right] \\
&= \Pr_{x \sim D}\left[ \begin{array}{l} t(x) = d_t[i + 2 :] \circ \gamma_{i+1} \circ d_h[i + 1] \circ \alpha_i \circ d_t[: i + 1](x) \wedge \\ t(x) = d_t[i + 1 :] \circ \gamma_i \circ d_h[: i + 1] \circ \alpha_0(x) \end{array} \right] \\
&= Pr[CR_{i+1} \wedge PR_i]
\end{aligned}
$$

by the assumption that all concrete components are invertible and that $\gamma_i = \alpha_i^{-1}$. The proof proceeds as before.

In particular, even with strong conditions such as these, we derive a bound which grows very weak as depth increases, and hence, we observe maintaining Axioms 1 and 3 in addition to Axioms 2 and 4 is essential for a reliable evaluation. See Appendix F for empirical evidence for this fact.

## F. Importance of Prefix Axioms: Empirical Evidence

We can demonstrate the importance of the prefix axioms by simulating errors in the interpretation of the 2-SAT model; specifically, every clause output by the first component of the abstract model is replaced by a randomly sampled clause 1% of the time (the slight difference in e.g. the epsilons for component and prefix equivalence for the first component are due to independent sampling); results are shown in Table 1.

Table 1: Comparison of results of experiments with synthetic error injected into the first component of the abstract model to original experimental results.

| Axiom | Noised first abstract component | | | Original abstract model | | |
|---|---|---|---|---|---|---|
| | Component 1 | Component 2 | Component 3 | Component 1 | Component 2 | Component 3 |
| Axiom 1: Prefix Equivalence | 0.0955 | 0.212 | 0.0315 | 0.0000374 | 0.182 | 0.0128 |
| Axiom 2: Component Equivalence | 0.0942 | 0.182 | 0.00433 | 0.0000374 | 0.182 | 0.00433 |
| Axiom 3: Prefix Replaceability | 0.0583 | 0.0312 | 0.0318 | 0.0418 | 0.0128 | 0.0128 |
| Axiom 4: Component Replaceability | 0.0581 | 0.0128 | 0.00433 | 0.0418 | 0.0128 | 0.00433 |

The evaluation of the componentwise axioms are identical to the original results except on the first component, as these axioms do not take error propagation into account. The prefix axioms, on the other hand, show the extent to which errors in the earlier components result in downstream errors.

Taking error propagation into account is particularly important when we consider deeper models. While we can derive worst-case compositional bounds from the componentwise results, this may not actually describe model behavior; in particular, as the number of components increases, the derived bound rapidly becomes meaningless (see Appendix E for more details). For example, in our case, the aggregation operation in the final component enables prefix equivalence to improve with depth; hence, restricting ourselves to the pure worst-case analysis derived from component-only analysis would suggest that our interpretation is far less reliable end-to-end than is actually the case.

## G. Neuron Interpretations for the 2-SAT Model

Tables 2 and 3 show the decision tree based and disjunction-only neuron interpretations, respectively.

## H. More Details on Mechanistic Interpretability Analysis of the Modular Addition Model

Pseudocode for the abstract model is shown in Listing 8. The concrete model is likewise broken into three corresponding components. The embedding matrix corresponds to `encoding_of_inputs`, the attention mechanism and the input layer of the MLP correspond to `sum_of_angles`, and the output layer of the MLP and the unembedding matrix correspond to `difference_of_angles_argmax`. The abstraction and concretization operators are linear maps learned using the continuous-valued abstract values (e.g. prior to rounding) and the corresponding concrete intermediate representations on the training set, using the original train/test split of Nanda et al. (2023a).

Note that, in Nanda et al. (2023a)'s paper, the third component is further decomposed into two pieces; specifically, the difference-of-angles identities and the summation (along with the argmax) steps are considered separate. The difference-of-angles identity is computed using the MLP output layer and the unembedding matrix while the summation step is computed via the unembedding matrix. As there is no canonical factorization of the unembedding matrix, there exists no natural decomposition of the concrete model into these two components—we analyze the composition of these two components instead.

As noted in Section 5, the discretization operators are necessary for compatibility with Axioms 1 and 2. Without rounding on the first component, we obtain epsilons of 1 for Axioms 1 and 2; epsilons for Axioms 3 and 4 are unaffected by rounding.

For the second component, we apply the equivalence class formulation described in Section 5; however, we strengthen the equivalence criterion by ensuring that equal samples affect neither concrete nor abstract model behavior. This was chosen to avoid the need to sample from the equivalence class on concretization or to select a canonical representative for each class; doing so would have been needed to correctly evaluate Axioms 3 and 4 without such a guarantee. When sampling in this way is feasible, an equivalence relation up to the abstract only model is sufficient. Pseudocode is given in Listing 9.

As for the first component, we obtain epsilons of 1 for Axioms 1 and 2 without applying the equivalence class mapping; by construction, the epsilons for Axioms 3 and 4 are unaffected by the equivalence class operation. Note that simple discretization operators do not suffice: we once again obtain an $\epsilon$ of 1 for Axioms 1 and 2 when rounding to a single decimal place. While strong results with the discretization described above show that the abstract and concrete representations of the output of the second component are equivalent up to their downstream impact on both the concrete and abstract models, results with simpler discretizations indicate that the the concrete representation does not represent the abstract state exactly.

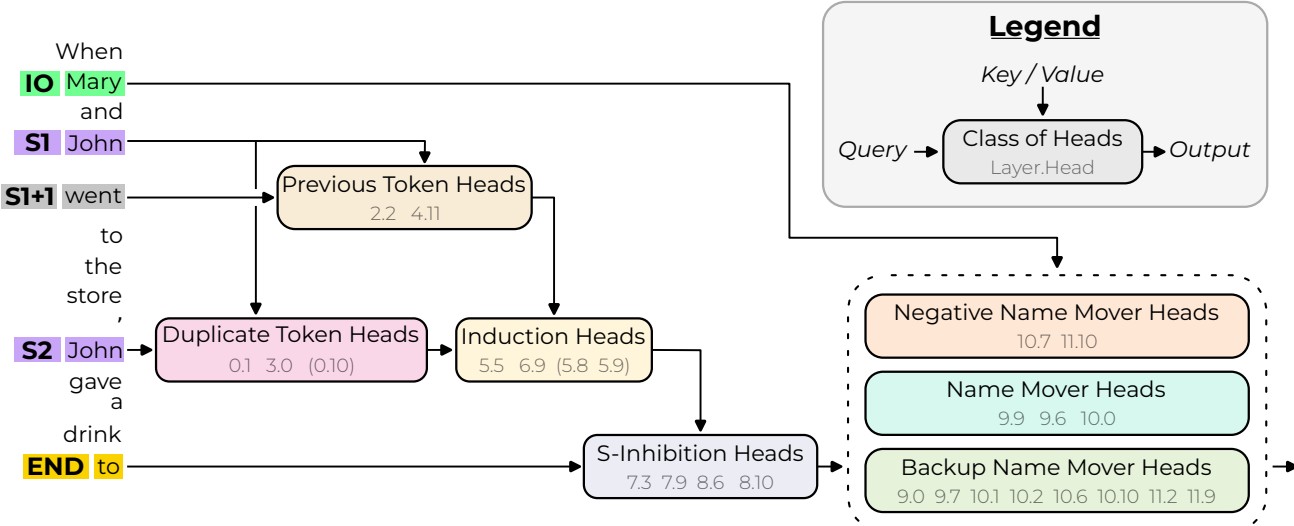

Figure 18: The IOI circuit, reproduced from Wang et al. (2023).

## I. A Sketch of Application to Circuits: Indirect Object Identification

As described in Appendix B.1, our axioms may be extended to evaluate circuits-type interpretations, or may be applied directly by linearization of the computational graph of the circuit.

To illustrate the process, we'll describe how the circuit for Indirect Object Identification (IOI) (Wang et al., 2023) may be evaluated. The IOI circuit consists of five broad categories of heads:

1. **Previous token heads**, which copy information from the prior token

2. **Duplicate token heads**, which identify whether there exists any prior duplicate of the current token

3. **Induction heads**, which serve the same function as duplicate token heads, mediated via the previous token heads

4. **S-inhibition heads**, which output a signal suppressing attention by name mover heads to duplicated names

5. **Name mover heads**, which copy names except those suppressed by the S-inhibition heads for output

Note that each of these attention heads computes a well-defined interpretable function which can be represented in our framework. Figure 18 shows the structure of the circuit. We'll now illustrate the process by which the circuit may be analyzed via the linearization technique from Appendix B.1.2 or the graph axioms from Appendix B.1.1. The circuit in Figure 18 cannot be directly evaluated, as it must be made isomorphic to a decomposition of the concrete model.

To do so, we will first expand the circuit in Figure 18 to a head-by-head graph. Next, for each block with an interpreted head, we construct dummy nodes in the interpretation which correspond to uninterpreted heads; as the interpretation claims that these are uninvolved in the IOI task, these nodes compute the identity function, a constant function or some other no-op which is independent of the IOI task.

Now, we must account for computations performed in uninterpreted blocks and in MLPs. We can express the computation performed by a block of a transformer as a simple computational graph: each interpreted head, and well as the dummy node corresponding to the uninterpreted heads, takes the incoming residual stream as an input and returns an update to the residual stream. These updates are added to the residual stream and processed by the remaining components, such as the MLP and any normalization, which produce the final residual output by the block. For simplicity, we combine any uninterpreted layers into this operation. These nodes, abstractly, update the program state from the potentially redundant input information. In this way, we can construct isomorphic concrete and abstract models from any circuit of this form.

We derive the concrete and abstract models shown in Figure 19. Note that structuring the circuit in this way allows us to clearly define and evaluate a hypothesis on how redundant copies of a given abstract concept are combined downstream.

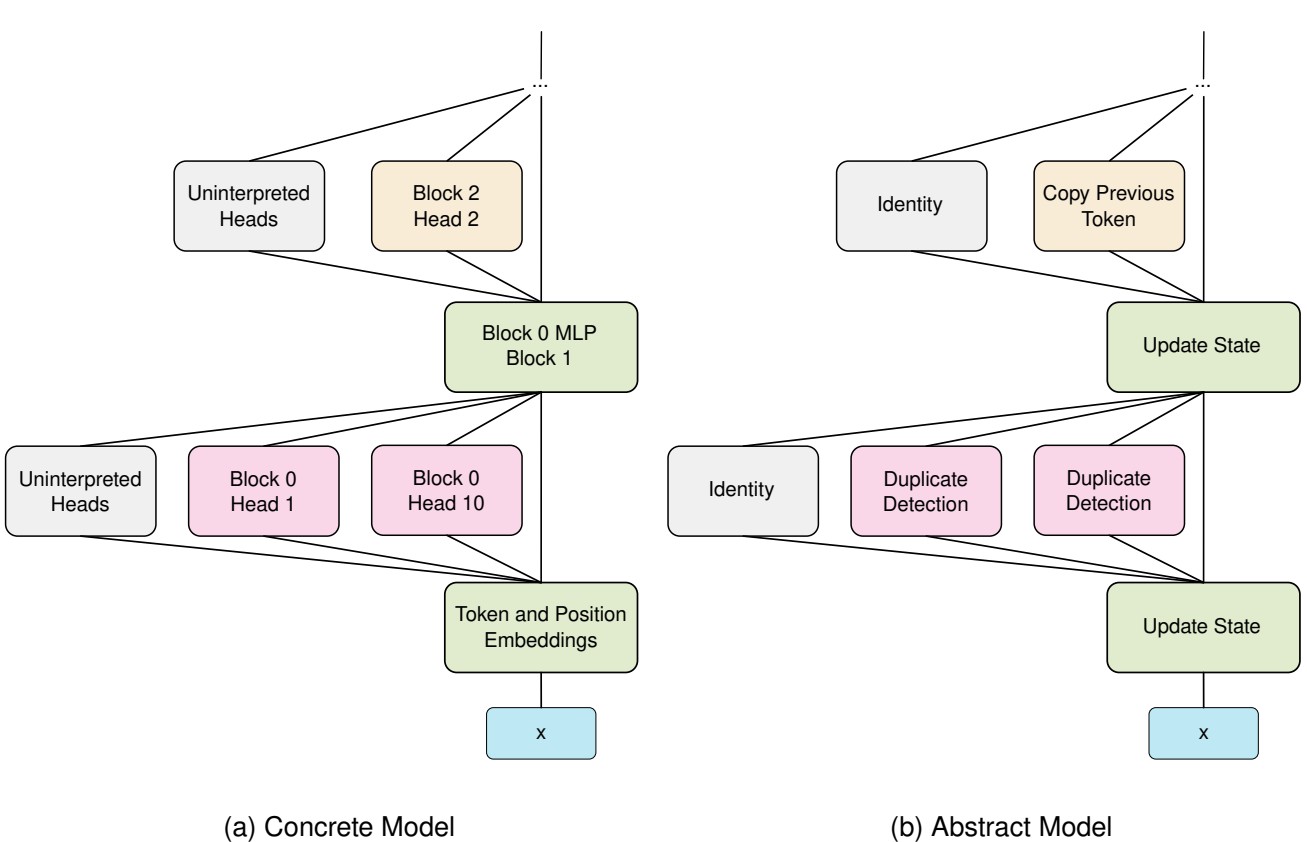

(a) Concrete Model                          (b) Abstract Model

Figure 19: Concrete and abstract models derived from the IOI circuit.

For example, the model might retain independent duplicate copies of the information, one copy per head, retain only the most recently-computed value, or aggregate redundant copies with some weighting.

At this point, we can apply either the linearization technique of Appendix B.1.2 or the generalized axioms of Appendix B.1.1 to evaluate the hypotheses. We do not fully evaluate IOI with our axioms here as Wang et al. (2023) leaves key components of the interpretation unspecified. In particular, the authors do not clearly state how the model combines the information produced by redundant heads performing the same function, and the authors do not conclusively state what the duplicate-token suppressing information output by the S-inhibition heads represents. For example, that information may be represented as the suppressed token itself, via either relative or absolute positions, or a combination of token and position information. A complete analysis with our axioms would enable the analyst to derive clear evidence for the correct hypotheses.

Table 2: General neuron interpretations derived by decision tree learning on neuron activations. A neuron activates if the corresponding condition in the second column holds.

| Neuron | Interpretation (General Case) |
|---|---|
| 10 | $\phi[TFFFF]$ |
| 29 | $\phi[FTTFF] \wedge \neg\phi[FFFFT] \wedge \neg\phi[TTTTT]$ |
| 36 | $\phi[TTFFF] \wedge \neg\phi[FTFTF]$ |
| 41 | $\phi[FFFFF]$ |
| 48 | $(\neg\phi[FTTFF] \wedge \phi[FTFFF]) \vee \phi[FTTFF]$ |
| 55 | $\phi[TTTFF]$ |
| 77 | $\phi[FFTTT]$ |
| 81 | $\phi[FTTTF]$ |
| 96 | $\phi[TFTTF] \wedge \neg\phi[FFTFF]$ |
| 150 | $\phi[FTFTF] \wedge \neg\phi[TTFFF]$ |
| 185 | $\phi[FFFTT]$ |
| 189 | $\phi[FTTTT] \wedge \neg\phi[TTTFT]$ |
| 195 | $\phi[TFFTF] \wedge \neg\phi[TTTTT]$ |
| 201 | $\phi[TTFFT] \wedge \neg\phi[TFTFF]$ |
| 224 | $\phi[FFFFT]$ |
| 231 | $\phi[FTFTT] \wedge \neg\phi[FFTTF]$ |
| 245 | $\phi[FTFFT]$ |
| 261 | $\phi[TTFTT]$ |
| 291 | $\phi[TFTTT]$ |
| 304 | $\phi[FFTTF] \wedge \neg\phi[TFTFF]$ |
| 317 | $\phi[TFFFT]$ |
| 326 | $\phi[TTTTF]$ |
| 334 | $\phi[FTTTF] \wedge \neg\phi[FFFTT]$ |
| 374 | $\phi[TTFTT] \wedge \neg\phi[TFTTF] \wedge \neg\phi[FFFFF]$ |
| 380 | $\phi[TFFTT]$ |
| 411 | $\phi[TTFFT]$ |
| 416 | $\phi[TTTFF]$ |
| 435 | $(\phi[TTFTF] \wedge \neg\phi[FTFFF]) \vee (\phi[TTFTF] \wedge \phi[FTFFF] \wedge \phi[TFFTF])$ |
| 450 | $\phi[FFTFT] \wedge \neg\phi[FTFFF]$ |
| 482 | $\phi[TTTTT] \wedge \neg\phi[FTTFT]$ |
| 490 | $(\phi[FTTFT] \wedge \neg\phi[TTTTT]) \vee (\phi[FTTFT] \wedge \phi[TTTTT] \wedge \neg\phi[TTTFT])$ |
| 492 | $\phi[TFTFT]$ |
| 495 | $\phi[FFFTF]$ |
| 499 | $(\phi[FFTFF] \wedge \neg\phi[TFTTF]) \vee (\phi[FFTFF] \wedge \phi[TFTTF] \wedge \phi[FFTTF])$ |

Table 3: Disjunction-only neuron interpretations derived by finding satisfying assignments that are correlated with a high average neuron activation value over the analysis training set. A neuron activates if the corresponding condition in the second column holds.

| Neuron | Interpretation (Disjunction-Only) |
|---|---|
| 10 | $\phi[TFFFF] \vee \phi[TFTFF] \vee \phi[TTFFF]$ |
| 29 | $\phi[FTTFF]$ |
| 36 | $\phi[TTFFF]$ |
| 41 | $\phi[FFFFF] \vee \phi[FTFFF]$ |
| 48 | $\phi[FTFFF] \vee \phi[FTTFF]$ |
| 55 | $\phi[TTTFF] \vee \phi[TTTFT]$ |
| 77 | $\phi[FFTTF] \vee \phi[FFTTT] \vee \phi[FTTTT]$ |
| 81 | $\phi[FTTTF] \vee \phi[FTTTT]$ |
| 96 | $\phi[TFTTF]$ |
| 150 | $\phi[FTFTF]$ |
| 185 | $\phi[FFFTT]$ |
| 189 | $\phi[FTTTT]$ |
| 195 | $\phi[TFFTF] \vee \phi[TFTTF] \vee \phi[TTFTF]$ |
| 201 | $\phi[TTFFT]$ |
| 224 | $\phi[FFFFT]$ |
| 231 | $\phi[FTFTT] \vee \phi[FTTTT]$ |
| 245 | $\phi[FTFFF] \vee \phi[FTFFT] \vee \phi[FTTFT]$ |
| 261 | $\phi[TTFTT]$ |
| 291 | $\phi[TFTTF] \vee \phi[TFTTT] \vee \phi[TTTTT]$ |
| 304 | $\phi[FFTTF]$ |
| 317 | $\phi[TFFFT]$ |
| 326 | $\phi[TTFTF] \vee \phi[TTTTF] \vee \phi[TTTTT]$ |
| 334 | $\phi[FTTTF]$ |
| 374 | $\phi[TTFTT]$ |
| 380 | $\phi[TFFTT]$ |
| 411 | $\phi[TTFFT] \vee \phi[TTTFT]$ |
| 416 | $\phi[TFTFF] \vee \phi[TTTFF]$ |
| 435 | $\phi[TTFTF]$ |
| 450 | $\phi[FFTFF] \vee \phi[FFTFT] \vee \phi[FTTFT]$ |
| 482 | $\phi[TTTTT]$ |
| 490 | $\phi[FTTFT]$ |
| 492 | $\phi[TFTFF] \vee \phi[TFTFT] \vee \phi[TTTFT]$ |
| 495 | $\phi[FFFTF]$ |
| 499 | $\phi[FFTFF]$ |

```
1  modulus = 113
2  key_freqs = [14, 35, 41, 42, 52]
3
4  cos_sin = tuple[list[float], list[float]]
5
6  def encoding_of_inputs(
7      a: int,
8      b: int,
9  ) -> tuple[cos_sin, cos_sin]:
10     cos_a = [round(cos(2 * f * pi * a / modulus)) for f in key_freqs]
11     cos_b = [round(cos(2 * f * pi * b / modulus)) for f in key_freqs]
12     sin_a = [round(sin(2 * f * pi * a / modulus)) for f in key_freqs]
13     sin_b = [round(sin(2 * f * pi * b / modulus)) for f in key_freqs]
14
15     return (cos_a, sin_a), (cos_b, sin_b)
16
17 def sum_of_angles(
18     components: tuple[cos_sin, cos_sin],
19 ) -> EquivalenceClass:
20     (cos_a, sin_a), (cos_b, sin_b) = components
21     cos_ab = [
22         ca * cb - sa * sb
23         for ca, sa, cb, sb in zip(cos_a, sin_a, cos_b, sin_b)
24     ]
25     sin_ab = [
26         sa * cb + ca * sb
27         for ca, sa, cb, sb in zip(cos_a, sin_a, cos_b, sin_b)
28     ]
29
30     return EquivalenceClass(cos_ab, sin_ab)
31
32 def difference_of_angles_argmax(angle_sums_class: EquivalenceClass) -> int:
33     cos_ab, sin_ab = extract_angle_sums(angle_sums_class)
34     cos_ab_minus_c = [
35         [
36             cab * cos(2 * f * pi * c / modulus) + sab * sin(2 * f * pi * c / modulus)
37             for cab, sab, f in zip(cos_ab, sin_ab, key_freqs)
38         ]
39         for c in range(modulus)
40     ]
41
42     return argmax(map(sum, cos_ab_minus_c))
43
44 def modular_addition(a: int, b: int) -> int:
45     return difference_of_angles_argmax(
46         sum_of_angles(
47             encoding_of_inputs(a, b)
48         )
49     )
```

Listing 8: Mechanistic interpretation of the modular arithmetic model

```
1  def equivalent(a: cos_sin, b: cos_sin) -> bool:
2    abstract_eq = abstract_components[3](a) == abstract_components[3](b)
3    concrete_eq
4      = concrete_components[3](gammas[2](a)) == concrete_components[3](gammas[2](b))
4    return abstract_eq and concrete_eq
```

Listing 9: Equivalence relation for second component of mechanistic interpretation of the modular arithmetic model

