# OpenReview forum: "Validating Mechanistic Interpretations: An Axiomatic Approach"
_ICML.cc/2025/Conference — ICML 2025 poster_

### Official Review · Reviewer_QoaE · 2025-02-25

**Overall Recommendation:** 2

**Summary:**

They propose a framework to quantify the effectiveness of a total decomposition of a model into sequential interpretable components. They find that this framework is able to validate (by showing a high probability in each of the equations of their axioms) explanations they create for the model components of a transformer trained to classify satisfiable 2-sat formulas. They also use their axioms to validate a previous interpretation (from [Nanda et. al. 2023 https://openreview.net/forum?id=9XFSbDPmdW]) for a transformer performing modular addition.

## update after rebuttal

I find that my opinion of the approach remains after the rebuttal period, and retain my initial score.

**Claims And Evidence:**

I find that they successfully demonstrate their method in toy examples. However, there are a few points that would be required to analyse their claims and methods in my opinion:
- Their axioms at face value require a full decomposition of a network into interpretable components. Since it is exceedingly unlikely that full model interpretations will exist for practical problems, one needs to interpret their axioms for sub-networks in practice; They go over how to re-interpret their axioms for these cases in Appendix B.1. I would argue that to sufficiently prove the utility of their framework, some practical example of existing foundation models should be experimentally analysed using their axioms as interpreted in B.1.
- Further, Their method is demonstrated, but not evaluated relative to anything else. For example one could design experiments where a model has a ground truth interpretation, and then show their method evaluates the correct interpretation in a better way than a false but plausible interpretation relative to some baselines.

**Essential References Not Discussed:**

Not to my knowledge.

**Experimental Designs Or Analyses:**

Yes, I checked over the details of the experiments in section 4 and 5 and found no issues.

**Methods And Evaluation Criteria:**

Their toy examples are appropriate, but more work and experiments would be required to evaluate their framework in practical foundation models. Also, their methods should be evaluated in their identification of good interpretations versus other baselines.

**Other Comments Or Suggestions:**

- I’m not sure if axiom is the right or best terminology for the inequalities called as such. A better description would be to treat those as “degrees of explainability”. I.E. the probability in the respective “axiom” quantifies the degree of explainability of the interpretation across prefix equivalence, prefix replaceability, etc …
- Typo 118 right: “functions need to be individually instantiated every mechanistic interpretation.”
- Since the computational graph is not just a sequence, how does the residual work with indexing and in all the axioms when switching from human and model components with alpha/gamma? Perhaps a sentence stating how to handle residual connections in general in the main text.
- Are the axioms for some fixed epsilon? Maybe denote them as Axiom i (\varepsilon) to be explicit.
- Be explicit when stating the axioms that the equality in the probability for each axiom is appropriate for the axioms when it does not take place in a high dimensional vector space, but rather the discrete (human-interpretable) output space or the human interpretable space. This is mentioned later in sentence at line 389 right when discretizing is necessary to evaluate the interpretation in section 5.
The discussion following this in line 418 right, highlights a general weakness of these axioms when the equality is evaluated in a high dimensional vector space, since discretization may be necessary to apply their axioms to a given model interpretation, as was the case for section 5. “[discretization] limits the ability of Axioms 1 and 2 to validate the internal behavior.”

**Other Strengths And Weaknesses:**

Weaknesses exist in terms of utility of their approach versus other methods for evaluating explainability of interpretation of model components.

Their interpretability analysis on the case study in section 4 is well done.

**Questions For Authors:**

- Why do these axioms in particular capture explainability well in a more appropriate way than some others? For example we could have formulated a single axiom where for any nondecreasing sequence i_n of indexes in [len(d)], we have the \varepsilon-replaceability of all components between i_{2k} , i_{2k+1}.
- Why the non-homogenous architecture in experiments, where the second layer has more attention heads?
- What is the loss function for training in experiments of section 4? Is it the usual next token prediction, or binary cross entropy on the sequence classification?

**Relation To Broader Scientific Literature:**

The broader context of this work is in the addition of their framework to the set of methods for evaluating the degree of validity of human-interpretations for model components.

**Theoretical Claims:**

Checked the arguments of appendix D and found no issues.

---

> ### Author Rebuttal · Authors · 2025-03-31
>
> Thank you for your helpful comments, we respond in detail below.
>
> ## Q1. More experiments are required to evaluate the framework by analyzing practical foundation models using the axioms from B.1
>
> We sketch how the circuit for IOI in GPT-2 [1] may be expressed and analyzed in our framework in the response to Q1 of reviewer LJQi. Evaluating circuits in our framework is straightforward given a fully-specified mechanistic interpretation. However, in practice, we observe that popular mechanistic interpretations such as the IOI circuit leave many key details unspecified. Circuit analyses of this type can be strengthened by evaluating using our framework as it forces the analysts to spell out all the details; we hope that such studies will be conducted in future work.
>
> [1] Wang et al. Interpretability in the Wild: A Circuit for Indirect Object Identification in GPT-2 Small. 2022.
>
> ## Q2. Framework should be evaluated in comparison to baseline evaluation frameworks
>
> We agree that evaluation of this type would be valuable, however, we note that we are the first to propose an axiomatic framework for the evaluation of mechanistic interpretations, presenting a natural and principled set of criteria; we believe that this already constitutes a valuable contribution to the community. We hope that our work inspires further conversation in this direction, and leave such an analysis to future work.
>
> ## Q3. How do the axioms handle residual connections?
>
> We can express residual connections via concrete functions which return the residual stream as part of a tuple of values. See the analysis of the 2-SAT model for an example: the second concrete component returns both the residual stream and the hidden layer of the MLP. Appendix B.1 discusses a variation of Axioms 1 through 4 which operate on nonlinear graphs, and in this case no change is necessary.
>
> In general, we may treat the residual connection as computing the identity function in the abstract model; we discuss this in more detail in the response to Q1 for reviewer LJQi.
>
> ## Q4. Are the axioms for some fixed $\epsilon$?
>
> Yes, that is correct. In definition 3.1, we define an interpretation as $\epsilon$-accurate when Axioms 1 through 4 are satisfied with that fixed value of $\epsilon$; we will clarify this.
>
> ## Q5. Axioms in high-dimensional vector spaces
>
> We do state that our axioms require that the abstract states be discrete (lines 128-130), however we will improve clarity.  No such restriction applies to the concrete model. While lines 410-420 illustrate an important tradeoff of discretization techniques, this is not a weakness of our axioms. Noting that our equivalence class formulation for discretization generalizes comparison up to a tolerance, any evaluation approach which considers equivalence of intermediate values (essential to have any hope at resolving the extensional equivalence problems of the type discussed in [2]) will observe the same behavior.
>
> Also see response to Q2 of reviewer zQST.
>
> [2] Scheurer et al. Practical pitfalls of causal scrubbing. AI Alignment Forum. 2023.
>
> ## Q6. Why do the axioms capture explainability better than other approaches?
>
> Firstly, our axioms define clear standards to evaluate. Approaches such as causal abstraction are overly broad, making it unclear to the analyst how their interpretation should be evaluated. If we view the equalities in our axioms as tests under causal interventions, we propose a specific and natural set of interventions which ensure that both internal and output behaviors of the interpretation and the concrete model match, and moreover that the corresponding abstract and concrete components are interchangeable.
>
> Secondly, we were motivated to formulate our axioms in a manner which captures validation techniques already used informally in the mechanistic interpretation community to evaluate interpretations.
>
> Finally, our axioms improve upon existing approaches by considering all of internal equivalence, output equivalence, and compositionality. We discuss the importance of compositional evaluation in appendices D and E. As shown by [2], evaluating only on outputs does not ensure that the intermediate values agree.
>
> We discuss these differences in more detail in section 3. We should note that we do not claim that our axioms cannot be improved upon; rather, our work represents a first step at formalizing validation of mechanistic interpretations. We hope that work in this direction continues and that our work inspires further improvements to evaluation techniques.
>
> Regarding the specific approach mentioned, as described above, techniques which neglect the equivalence of internal values invite the problem of only establishing extensional equivalence [2].
>
> ## Q7. Why the non-homogeneous architecture?
>
> Our goal was to identify the simplest architecture with high accuracy, which we observed with this architecture.
>
> ## Q8. What is the loss function?
>
> We use a next-token prediction objective.

---

> > ### Comment · Reviewer_QoaE · 2025-04-02
> >
> > >... we observe that popular mechanistic interpretations such as the IOI circuit leave many key details unspecified. Circuit analyses of this type can be strengthened by evaluating using our framework as it forces the analysts to spell out all the details …
> >
> > I don’t think that your axiomatization helped, or was required, to see that components were underspecified in the incomplete interpretation in this example.
> >
> > I do not think computing epsilon values for an interpretation in a vacuum has power in showing the validity/utility of your approach to “demonstrate the applicability of these axioms for validating mechanistic interpretations”. Furthermore, to properly contrast between two interpretation methods (eg like the 2-sat experiment aimed to do), one needs to check that a deeper causal analysis agrees with which of the two interpretations has lower epsilon values, otherwise there is the tautology of evaluating the axioms in a vacuum.
> >
> > In the 2-sat experiment, the epsilon values of different axioms even disagree: In the second block hidden layer, some axioms rank “decision tree” better, and some prefer “disjunction only”.
> >
> > As someone who is explicitly looking for ways to compare the quality of different interpretations for the same model component, I am not convinced that I should use your approach, without empirical justification of the distinguishing power of these axioms.

---

> > > ### Author Response · Authors · 2025-04-03
> > >
> > > > I don’t think that your axiomatization helped, or was required, to see that components were underspecified in the incomplete interpretation in this example.
> > >
> > > We do not argue that our axiomatization is necessary to identify that components are underspecified. The authors of [1] specify as such in their appendix. Our point is that as our axioms require fully-specified interpretations, evaluation with them would avoid such issues in the first place.
> > >
> > > > I do not think computing epsilon values for an interpretation in a vacuum has power in showing the validity/utility of your approach to “demonstrate the applicability of these axioms for validating mechanistic interpretations.”
> > >
> > > We should note that $\epsilon$ describes the probability that the behavior of the concrete and abstract models differ, and is meaningful in itself. We may view $\epsilon$ as a form of error, in the sense that $1 - \epsilon$ is the accuracy with which the abstract model or intervened concrete model predicts the behavior of the original concrete model. In our experiments, we observe values of $\epsilon$ in the full range of 0 to 1. We observe values of up to 1 for bad interpretations, as well as values of 0 for interpretations which perfectly match the behavior of the model, indicating that our axioms do indeed separate valid from invalid hypotheses.
> > >
> > > Moreover, by defining natural and desirable properties which any valid mechanistic interpretation should satisfy, our axioms not only formalize the extant informal practices for validating mechanistic interpretations used by the community but also highlight the weaknesses of the existing practices. For instance, our 2-SAT analysis in Appendix E illustrates the importance of compositional evaluation as imposed by the prefix axioms—a validation step that existing mechanistic interpretability analyses lack.
> > >
> > > > Furthermore, to properly contrast between two interpretation methods (eg like the 2-sat experiment aimed to do), one needs to check that a deeper causal analysis agrees with which of the two interpretations has lower epsilon values, otherwise there is the tautology of evaluating the axioms in a vacuum.
> > >
> > > > … I am not convinced that I should use your approach, without empirical justification of the distinguishing power of these axioms.
> > >
> > > We should note that our replaceability axioms *already express* such a causal analysis, of a form similar to those used in existing mechanistic interpretability works. In particular, they explicitly test the effect of intervention on the concrete state as a function of the abstract state.
> > >
> > > While we agree with the reviewer that a thorough evaluation comparing the distinguishing power of the axioms with that of other approaches would be useful, we emphasize that our work is the first of its kind and can spur further efforts at formalizing mechanistic interpretability. We believe such a formalization is valuable in its own right.
> > >
> > > > In the 2-sat experiment, the epsilon values of different axioms even disagree: In the second block hidden layer, some axioms rank “decision tree” better, and some prefer “disjunction only”.
> > >
> > > This is not a flaw of our axioms, and is in fact a desirable property. This is why an interpretation is $\epsilon$-accurate only when it is $\epsilon$-accurate in terms of *all* axioms. If all axioms agreed, they would be testing equivalent properties, and only one would be necessary.

---

### Official Review · Reviewer_zQST · 2025-03-13

**Overall Recommendation:** 2

**Summary:**

This paper introduces a set of axioms aimed at formalizing mechanistic interpretability for neural networks, inspired by abstract interpretation concepts in program analysis. The authors define mechanistic interpretations as human-interpretable programs that approximately replicate the computations of neural networks. To validate such interpretations, they propose six axioms that focus on input-output equivalence and component-level fidelity. The paper applies these axioms in two case studies: a previously known transformer-based modular arithmetic model and a novel transformer trained to solve the 2-SAT problem. Empirical analysis confirms the axioms can quantitatively validate mechanistic interpretations, contributing a structured evaluation framework.

**Claims And Evidence:**

The main claims of formalizing and empirically validating mechanistic interpretations through axioms are reasonably supported. However, critical claims about the general applicability and scalability of the proposed axiomatic framework beyond simplistic algorithmic tasks lack clear and convincing evidence. While results on toy examples (2-SAT, modular addition) are encouraging, the absence of larger or practically relevant neural network benchmarks diminishes the robustness of broader claims.

**Essential References Not Discussed:**

The paper neglects discussion of broader interpretability frameworks beyond the program-analysis-inspired perspective. Moreover, broader causal interpretation literature could have further contextualized and strengthened the motivation behind formal axiomatic approaches.

- Sundararajan, Mukund, Ankur Taly, and Qiqi Yan. "Axiomatic attribution for deep networks." International conference on machine learning. PMLR, 2017.
- Mueller, Aaron, et al. "The quest for the right mediator: A history, survey, and theoretical grounding of causal interpretability." arXiv preprint arXiv:2408.01416 (2024).

**Experimental Designs Or Analyses:**

The experimental validation provided is sound but limited. The analyses (attention patterns, decision tree extraction for neuron activations) for the chosen examples are adequate, though they only apply to very simple Transformer architectures trained on straightforward tasks. The robustness of these axioms when applied to larger and more complex datasets, or real-world tasks, is unclear. More diverse experimental setups would substantially strengthen the paper.

**Methods And Evaluation Criteria:**

The proposed axioms and evaluation methodology make conceptual sense for assessing mechanistic interpretations on small-scale, algorithmically transparent models. However, the chosen tasks—modular arithmetic and 2-SAT problems—are relatively trivial compared to practical neural network applications, limiting the generalizability and real-world relevance of the proposed evaluation criteria.

**Other Comments Or Suggestions:**

None.

**Other Strengths And Weaknesses:**

The paper's strength lies in providing clear formalization and axioms for validating mechanistic interpretations, filling an important gap in the literature. The theoretical clarity and detailed empirical evaluation on chosen tasks are commendable.

However, the primary weaknesses include:

- Limited practical relevance due to simplistic and overly specialized choice of tasks (2-SAT and modular arithmetic).

- Lack of demonstration or discussion on scalability to real-world, complex neural networks and tasks.

- The approach might become less meaningful or too restrictive for networks with significantly richer internal structures or continuous-valued intermediate states.

**Questions For Authors:**

- How does the proposed axiomatic approach scale to larger, real-world models, particularly when intermediate states are high-dimensional continuous vectors rather than discrete symbols?

- Could you provide concrete guidance on determining appropriate abstraction functions \alpha and concretization functions \gamma in realistic settings with large neural networks?

**Relation To Broader Scientific Literature:**

The paper situates itself effectively within existing literature, clearly distinguishing its contribution from related approaches such as causal abstraction and causal scrubbing. It successfully positions its axiomatic framework as complementary and extending prior works by emphasizing compositional evaluation.

**Theoretical Claims:**

The theoretical claims are clearly presented, and definitions of the axioms appear sound. I reviewed the definitions and the axiomatic formulations carefully. While the theoretical framing seems correct, no significant issues emerged. However, the theory primarily covers simple computational models, and extending it theoretically or empirically to more complex and realistic architectures would enhance its value significantly.

---

> ### Author Rebuttal · Authors · 2025-03-31
>
> Thank you for your thoughtful comments, we respond in detail below.
>
> ## Q1. Case studies are too simple; it is unclear whether the approach is applicable to larger models
>
> We note that our framework is already compatible with larger models and more complex architectures, and we emphasize that our main contribution is our evaluation framework (i.e., axioms) for validating mechanistic interpretations and not techniques to derive mechanistic interpretations. Hence, any increased difficulty of deriving mechanistic interpretations on larger models does not affect the applicability of our approach. See our response to Q1 by reviewer LJQi for more details.
>
> ## Q2. How does the proposed axiomatic approach scale to larger, real-world models, particularly when intermediate states are high-dimensional continuous vectors rather than discrete symbols?
>
> As noted above, our approach is compatible with larger models. We note that the intermediate states of the concrete model are, in all cases (including for 2-SAT and modular arithmetic models), high dimensional continuous-valued vectors. Our requirement that the intermediate states be discrete is only for the abstract model. In particular, we require this as we agree that mechanistic interpretations cannot hope to achieve equality of high-dimensional continuous vectors in Axioms 1 and 2.
>
> However, we emphasize that our requirement is natural for useful mechanistic interpretations, in particular, as any interpretations which operate on high dimensional continuous vectors will not generally be human-interpretable and hence will be of limited utility.
>
> ## Q3. Essential references not discussed
>
> Thank you for the references, we will incorporate them into the paper. We should note that while [1] proposes an axiomatic framework for interpretability, it considers input interpretability only. [2] is primarily a survey of the architectural components interpreted and techniques such as probing for identifying the representations of abstract values, and reflects the class of concrete programs which $\lambda_T$ must represent.
>
> [1] Sundararajan, Mukund, Ankur Taly, and Qiqi Yan. "Axiomatic attribution for deep networks." International conference on machine learning. PMLR, 2017.
> [2] Mueller, Aaron, et al. "The quest for the right mediator: A history, survey, and theoretical grounding of causal interpretability." arXiv preprint arXiv:2408.01416 (2024).

---

### Official Review · Reviewer_RksT · 2025-03-14

**Overall Recommendation:** 5

**Summary:**

This paper is a first effort to draw a parallel between mechanistic interpretability and abstract interpretation from programming language theory. This is a natural analogy, and its very exciting to see contact between these two areas. The authors introduce four axioms for how an abstraction interpretation of a neural network should operate. Then, they present two case studies. In the first, they train a two layer transformer on SAT solving and then develop a mechanistic interpretation by looking at attention heads and MLP activations. In the second, they reevaluate the modular arithmetic algorithm presented in Nanda (2022) under their framework. Both analyses are successful.

**Claims And Evidence:**

There are some claims about connections to causal scrubbing/abstraction that I would be curious to hear more about. What are the issues that come from only adhering to some of your axioms, and what are some precise examples where this approach differs from causal abstraction? Maybe something you could add something about this to your thorough appendix!

**Essential References Not Discussed:**

You should check out this paper:

https://arxiv.org/pdf/2301.04709

Which goes much further into the causal abstraction theory and connection to mechanistic interpretability. In particular, I would be curious what you make of Remark 35 given that you claim in this paper that causal abstraction lakes the concretization and abstraction operations.

I think you would find it deepens the connections to this work greatly!

**Experimental Designs Or Analyses:**

The experimental design seems very good to me. Easy to understand and well-written.

**Methods And Evaluation Criteria:**

I would say that the weakest part of the paper is that you don't include baseline evaluations that help contextualize your error terms. Sure, some of the error terms look really low, but without context new metrics are very difficult to understand! How about is the larger error you see in the MLP for the SAT solving mechanistic interpretation?

Is there some evaluation metric that is more similar to interchange intervention accuracy from causal abstraction stuff? Getting an impression of how much of the model behavior your mechanistic interpretation matches would be great!

**Other Comments Or Suggestions:**

N/A

**Other Strengths And Weaknesses:**

This paper is an original take on mechanistic interpretability and brings over exciting new tools and ideas from programming language theory. The experiments are simple, but they clearly demonstrate the relevant concepts.

The biggest weakness of this paper is that its evaluated on toy models, but I would say the primary contribution is the new framework and concepts that they are introducing to mechanistic interpretability.

**Questions For Authors:**

Could you give an intervention-based definition for the axioms you defined? Or maybe just explain whether/how interventions are being performed in your experiments on transformers? I think that interventions are being performed for each of the axiom verifications, but it would be cool to have it spelled out

**Relation To Broader Scientific Literature:**

Mechanistic interpretability is in need of more theoretical work that grounds out the field in existing theoretical frameworks. Prior work looks to formal theories of causality, but this work begins to build a bridge to program language theory which is a rich and well-studied field with formal foundations that mechanistic interpretability could use!

**Theoretical Claims:**

N/A

---

> ### Author Rebuttal · Authors · 2025-03-31
>
> Thank you for your detailed comments, and for your strong support for our work! We respond in detail below.
>
> ## Q1. What happens when only some of the axioms are adhered to?
>
> In appendices D and E, we include a discussion of what happens when we consider component axioms (Axioms 2 and 4) alone; in this case, we do not consider compositionality of the interpretation. Likewise, considering replaceability alone invites the problem of the interpretation being extensionally equivalent (equivalent in terms of input-output behavior) to the model under analysis but lacking intensional equivalence (equivalence in terms of internal structure) [1].
>
> [1] Scheurer et al. Practical pitfalls of causal scrubbing. AI Alignment Forum. 2023.
>
> ## Q2. How should the $\epsilon$ values be contextualized?
>
> We may view the $\epsilon$ as a form of error, in the sense that $1 - \epsilon$ is the accuracy with which the abstract model or intervened concrete model predicts the behavior of the original concrete model. In our experiments, we observe values of $\epsilon$ in the full range of 0 to 1. We observe values of up to 1 for bad interpretations, as well as values of 0 for interpretations which perfectly match the behavior of the model (the only reason the paper does not report any $\epsilon$ values of 0 is because we compute Clopper-Pearson confidence intervals). In this sense, $\epsilon$ describes the probability that the behavior of the concrete and abstract models differ, and is meaningful in itself. We should add that $\epsilon$ is, in addition, useful for relative comparisons, identifying which interpretation better matches the underlying behavior of the model.
>
> ## Q3. Relationship of our framework with causal abstraction and causal interventions
>
> We will make a note about Remark 35 of [2], thank you for the suggestion! However, it is important to note that $\tau^{-1}$ may not always be a feasible choice for concretization. In particular, $\tau$ will not be invertible in general, and hence this necessitates application of set semantics, which may be infeasible in the concrete domain, and hence for replaceability. In addition, we should note that it is not necessarily the case that abstract interpretations align individual high level with individual low level variables.
>
> While our axioms do not exactly follow the structure of interchange intervention (in particular, the interventions are not derived by substituting intermediate states from different inputs), they may be regarded as performing a specific class of causal interventions, which are soft interventions of the type described in [2].
>
> The replaceability axioms may be viewed as interventions on the original concrete model. If we consider component replaceability and $x_i$ is the input to the $i^{\text{th}}$ concrete component, the intervention performed in component replaceability replaces the constraint $x_{i+1} = d_t\[i\](x_i)$ with the constraint $x_{i+1} = \gamma_i \circ d_h\[i\] \circ \alpha_{i-1} x_i$. The equivalence axioms may be defined as analogous soft interventions on the corresponding abstracted prefixes of the concrete model.
>
> From that perspective, our axioms present a principled choice for a standardized set of causal interventions, which we believe is very valuable to the mechanistic interpretability community. In particular, while frameworks such as that of [2] are very broad, that breadth means that there is no clear standard choice for evaluation, hence causal abstraction analyses continue to emphasize interchange intervention accuracy, which does not directly evaluate the equivalence of internal representations.
>
> [2] Geiger et al. Causal Abstraction: A Theoretical Foundation for Mechanistic Interpretability. 2024.
>
> ## Q4. The framework is evaluated only on toy models.
>
> We do not intend the case studies presented to serve as an evaluation of our framework, but as an illustration of how it may be applied. Our framework can indeed be used to validate interpretations of larger models; see the response to Q1 of reviewer LJQi for more details. As the reviewer notes, our primary contribution is the new framework for evaluation of mechanistic interpretations and not particular techniques to derive them.

---

### Official Review · Reviewer_LJQi · 2025-03-14

**Overall Recommendation:** 3

**Summary:**

The paper introduces a formal framework for assessing mechanistic interpretations of neural networks. The authors propose a set of axioms inspired by abstract interpretation from program analysis to systematically evaluate whether a given mechanistic interpretation accurately captures a model’s internal computations. The proposed framework emphasizes compositionality, requiring that each step in the interpretation aligns with the corresponding computations in the network. The axioms are validated through case studies on Transformer-based models trained to solve the modular addition and the 2-SAT problem. Experimental results demonstrate that the axioms provide a structured way to evaluate interpretability quantitatively.

**Claims And Evidence:**

The major claims made by the paper are:
1. Mechanistic interpretability can be rigorously defined through a set of axioms that ensure both input-output fidelity and internal component alignment.
2. Valid interpretations should respect compositionality, meaning that replacing parts of a model with their interpreted counterparts should minimally affect outputs.
3. The proposed axioms can be empirically validated using statistical tests.

The claims are supported through experimental validation. The case studies show that the axioms can characterize valid mechanistic interpretations.

**Essential References Not Discussed:**

The paper discusses the most relevant work in mechanistic interpretability.

**Experimental Designs Or Analyses:**

The experimental setup is well-structured and supports the claims:
1. The 2-SAT model’s interpretation is carefully analyzed using attention patterns, decision trees, and abstraction-concretization mappings.
2. The modular addition model’s interpretation is validated through discretization techniques and statistical checks.
3. The impact of different abstraction choices (e.g., disjunction-only vs. full decision tree interpretations) is explored, demonstrating trade-offs in interpretability fidelity.

**Methods And Evaluation Criteria:**

The evaluation methods used in the paper are reasonable and align with the goal of assessing mechanistic interpretability. The authors apply their framework to two models, including an existing modular addition Transformer and a novel 2-SAT-solving Transformer. These evaluation strategies provide the empirical foundation for the proposed axiomatic approach.

**Other Comments Or Suggestions:**

N/A.

**Other Strengths And Weaknesses:**

Strengths:
1. The paper is well-written and presents a clear, formal framework for mechanistic interpretability.
2. The proposed axioms provide a structured way to evaluate interpretations, filling a gap in prior work.
3. The experimental validation is thorough, with strong empirical support for the claims.

Weaknesses:
1. The method primarily evaluates small-scale models trained on synthetic tasks (e.g., modular arithmetic and 2-SAT). It remains unclear how well this approach generalizes to larger, practical models.
2. Following 1, the axiomatic framework may require significant manual effort when applied to real-world models.

**Questions For Authors:**

1. How sensitive are the axioms to the choice of abstraction and concretization functions?
2. Is the proposed axioms sufficient to validate mechanistic interpretation?

**Relation To Broader Scientific Literature:**

The work extends prior research on mechanistic interpretability by providing a formal framework.

**Theoretical Claims:**

The paper argues that mechanistic interpretations can be formally validated using an axiomatic approach. The theoretical framework is sound, and the derivations appear correct.

---

> ### Author Rebuttal · Authors · 2025-03-31
>
> Thank you for your thoughtful comments, we respond in detail below.
>
> ## Q1. It is unclear how well this approach generalizes to larger models
>
> We emphasize that our key contribution is our framework (i.e., axioms) for evaluation of mechanistic interpretations and not particular techniques to derive them. Hence, while deriving/finding mechanistic interpretations is more difficult on larger models, it does not impact the applicability of our approach. As an illustration, consider the application of our framework to IOI [1].
>
> The IOI circuit for GPT-2, at a high level, identifies the indirect object by identifying all names in a sentence, filtering out any duplicated names, and selecting the remaining name.
>
> The analysis in [1]. is at the level of attention heads, and consists of five broad categories of heads:
>
> 1. **Previous token heads**, which copy information from the prior token
> 2. **Duplicate token heads**, which identify whether there exists any prior duplicate of the current token
> 3. **Induction heads**, which serve the same function as duplicate token heads, mediated via the previous token heads
> 4. **S-inhibition heads**, which output a signal suppressing attention by name mover heads to duplicated names
> 5. **Name mover heads**, which copy names except those suppressed by the S-inhibition heads for output
>
> Each of these attention heads computes a well-defined human-interpretable function and are representable in our framework; each may be modeled as an independent node in the computational graph. Each depends on the residual stream flowing into the corresponding layer; the subsequent residual stream is then a function of the preceding residual stream and all interpreted attention heads. As only attention heads are interpreted, the remainder of a block is an identity function in the abstract model. In this way, we can construct isomorphic concrete and abstract computational graphs for arbitrary circuits.
>
> Appendix B.1 describes how the resulting computational graphs may be linearized, allowing the application of axioms 1 through 4, and how to extend our axioms to operate natively on nonlinear computational graphs.
>
> Moreover, our framework provides a principled approach for evaluating a number of key questions left by [1] for future work. In particular, the authors of [1] hypothesize that S-inhibition heads output relative, and not absolute positions of duplicated names, but do not clearly demonstrate that this is the case. Applying our axioms to the corresponding abstract models would permit the analyst to identify the better hypothesis.
>
> However, we again emphasize that our focus is on the evaluation framework, and not methods for deriving mechanistic interpretations. Thus, while the analysis of GPT-2 Small and other larger models is compatible with our axioms, conducting such an analysis and deriving mechanistic interpretations is out of scope for our work.
>
> [1] Wang et al. Interpretability in the Wild: A Circuit for Indirect Object Identification in GPT-2 Small. 2022.
>
> ## Q2. Significant manual effort may be necessary
>
> We agree that significant effort is necessary for the analyst to derive a mechanistic interpretation, but it should be noted that mechanistic interpretability is in general a time-consuming and highly manual process. While automated approaches to deriving interpretations are an interesting direction of research in their own right, our work is about evaluation of derived mechanistic interpretations and not about techniques to derive an interpretation.
>
> ## Q3. How sensitive are the axioms to the choice of abstraction and concretization functions?
>
> The axioms may be sensitive to this choice, but we should note that these are essential components of the interpretation; interpretations which claim that the same abstract function is computed over different representations are not the same, and hence their evaluations should not be either. It is the analyst's responsibility to choose these functions appropriately, and our axioms are useful for identifying which choice more closely matches the behavior of the model.
>
> ## Q4. Are the proposed axioms sufficient to validate mechanistic interpretations?
>
> Yes, our axioms are sufficient, presenting a strong set of criteria to ensure that the behavior of the concrete model and the claimed mechanistic interpretation are interchangeable and behave in the same way. We recognize that stronger axioms are possible, and, in particular, we propose two additional axioms, Axiom 5 and 6 in the appendix. These strengthen the evaluation, but are difficult to operationalize, and we believe that Axioms 1 through 4, as presented in the main body, already constitute an effective evaluation valuable to the mechanistic interpretability community. Our work represents a first step in the direction of formally validating mechanistic interpretations. We hope that our work inspires a conversation in the community and leads to further improvements to evaluation techniques.

---

### Decision · Program_Chairs · 2025-05-01

**Decision:**

Accept (poster)

**Comment:**

In the context of neural network interpretability, this paper focuses on evaluating mechanistic interpretations that aim to capture, in an intelligible way, the computations performed by predictive models. Drawing inspiration from the concept of abstract interpretation found in program analysis literature, the authors adopt an axiomatic approach for assessing mechanistic interpretations. Specifically, they introduce four axioms to validate the capacity of a given mechanistic interpretation to approximately capture the semantics and compositional structure of the predictive model. The applicability of this axiomatic validation approach is examined in two case studies involving transformer architectures: one involving satisfiability of 2-CNF formulae and another on modular addition.

While this study has received mixed reviews, ranging from "Strong Accept" to "Weak Reject," reviewers generally agree on the novelty and flexibility of the axiomatic validation approach. However, there are concerns regarding the current evaluation on toy models and its potential scalability to more complex and realistic architectures. Additionally, some reviewers noted a lack of detailed comparison with causal abstraction approaches. Considering the reviews and the authors' responses during the rebuttal, the strengths of this paper—particularly the originality of the proposed framework—outweigh its weaknesses. Accordingly, I recommend a "Weak Accept" for the paper.

In a revised version, I suggest including a detailed discussion on how well this axiomatic validation approach can be extended to larger predictive models used in practice. Adding more case studies (deferred to the Appendix) could indeed further demonstrate the practical applicability of this approach.